# Observation of an electronic order along [110] direction in FeSe

Kunliang Bu[1], Wenhao Zhang[1], Ying Fei[1], Yuan Zheng[1], Fangzhou Ai [1], Zongxiu Wu[1], Qisi Wang[2], Hongliang Wo[2], Jun Zhao [2,3] & Yi Yin [1,3✉]

Multiple ordered states have been observed in unconventional superconductors. Here, we apply scanning tunneling microscopy to probe the intrinsic ordered states in FeSe, the structurally simplest iron-based superconductor. Besides the well-known nematic order along [100] direction, we observe a checkerboard charge order in the iron lattice, which we name a [110] electronic order in FeSe. The [110] electronic order is robust at 77 K, accompanied with the rather weak [100] nematic order. At 4.5 K, The [100] nematic order is enhanced, while the [110] electronic order forms domains with reduced correlation length. In addition, the collective [110] order is gaped around [−40, 40] meV at 4.5 K. The observation of this exotic electronic order may shed new light on the origin of the ordered states in FeSe.

[1] Zhejiang Province Key Laboratory of Quantum Technology and Device, Department of Physics, Zhejiang University, Hangzhou, China. [2] State Key Laboratory of Surface Physics and Department of Physics, Fudan University, Shanghai, China. [3] Collaborative Innovation Center of Advanced Microstructures, Nanjing University, Nanjing, China. ✉email: yiyin@zju.edu.cn

The mystery of high-temperature superconductivity comes from complicated ordered states, among which the electronic order and spin order have been extensively studied. For example, the electronic nematic order has attracted a lot of interests in both iron-based superconductors[1–5] and cuprate superconductors[6–9]. For the most common electronic nematic order, a four-fold rotational symmetry is broken into a two-fold symmetry along the [100] direction (Cu–O–Cu or nearest Fe–Fe direction) of the superconducting layer[1–9]. Recently, electronic orders along different directions have attracted researchers' attention[10–14]. To understand the relationship between different ordered states is of fundamental interest in the field of superconductivity.

FeSe is the structurally simplest iron-based superconductor, with the bulk FeSe composed of stacked Fe–Se–Fe layers[15]. Despite a low $T_c \sim 9$ K in the bulk FeSe, the superconducting critical temperature can be greatly boosted by carrier doping[16–19], by external pressure[20–23], or in a single layer FeSe/SrTiO$_3$ film[24,25]. Electronic nematicity is a hot topic in FeSe[26–29]. In both bulk FeSe and FeSe films, electronic nematicity along [100] direction has been intensively studied, by angle-resolved photoemission spectroscopy (ARPES)[30–36], scanning tunneling spectroscopy (STS)[37–43], nuclear magnetic resonance (NMR)[44–49], inelastic neutron scattering (INS)[50–52], Raman scattering[53–55] and transport measurement[56]. In iron-based superconductors, the [100] nematicity is usually accompanied with a collinear antiferromagnetic (AFM) order. However, this long-range magnetic order is absent in bulk FeSe, raising a question whether other ordered states coexist and compete with the [100] order[11,14]. On the other hand, the origin of [100] nematicity in FeSe is far from understood, with both orbital order[30,31,44–46] and spin fluctuations[47,48,50] proposed as potential mechanisms. The emergence of other electronic states may help disentangle the remaining debate about the origin of electronic nematicity.

Scanning tunneling microscopy (STM) is an atomic-resolved technique to detect the local electronic density of states (LDOS) of materials. In previous STM work about FeSe, the [100] nematicity is determined by a unidirectional dark stripe straddling a dumbbell defect[39,57]. The dumbbell defect serves as a pinning center to present electronic nematicity[40]. Different types of defects may reflect distinct parts of the electronic structure[58,59]. No symmetry breaking has been reported around the Se vacancy in FeSe. Electronic nematicity observed upon dumbbell defects in STM experiments thus relies on the type of defects. The intrinsic ordered states in a clean area can also be directly probed by STM, as the application in cuprate superconductors[6–8]. Here in this paper, we apply low-temperature STS to explore the intrinsic ordered states in bulk FeSe. A novel checkerboard charge order is observed in the iron lattice, which we name a [110] electronic order in FeSe. At liquid nitrogen temperature around 77 K, the robust [110] electronic order is accompanied with a weak [100] nematic order. At liquid helium temperature around 4.5 K, the [100] nematic order is enhanced, while small-sized domains appear in the pattern of the [110] electronic order. At 4.5 K, the [110] electronic order is also gapped within the energy range around [−40, 40] meV, a behavior reminiscent of the gapped Néel spin fluctuations in previous INS experiment[51]. The trade-off effect implies a competition between the [100] and [110] electronic orders.

## Results

**Intrinsic electronic orders at 77 K.** In the bulk crystal, each single layer of FeSe is composed of a square lattice of Fe atoms sandwiched between two planes of Se lattice. Figure 1a shows a schematic top view of a single layer of FeSe. We classify Fe atoms into two groups, Fe$_1$ and Fe$_2$, depending on its direction of bonding with Se atoms in the top plane. The $a/b$ $(x/y)$ axes are labeled along two perpendicular directions of the Fe (Se) lattice. The notation [100] is defined based on the $a/b$ axes of nominal Fe lattice. Below a critical temperature $T_s \sim 90$ K, a structural transition from tetragonal to orthorhombic lattice induces a small difference between lattice constants along $a$ and $b$ axes. The bulk FeSe is cleaved between two adjacent FeSe layers, with an electrically neutral Se plane exposed for the STM measurement. Figure 1b is an atomic-resolved topographic image of exposed Se plane, with each bright spot representing a Se atom. The interatomic spacing is measured to be $a_0 = 0.37$ nm, consistent with the X-ray diffraction result[60]. The difference between lattice constants along $a$ and $b$ axes is too small (5‰) to be resolved in an STM experiment[22]. The square lattice of top Se atoms is compatible with the black diamond in Fig. 1a. In Fig. 1b, some typical defects can be observed, including bright dumbbells, dark Se vacancies, and less dark defects at inner-layer Se sites. The zoom-in images of these defects are shown in Fig. 1c. Each dumbbell defect is centered at the Fe site, which is most possibly the Fe vacancy[61]. Two bright spots of the dumbbell are at the Se site. The dumbbell directions are distributed randomly along two perpendicular Se–Se directions (Fig. 1c, red squares). The four-fold symmetric defect at inner-layer Se site is speculated to be the inner-layer Se vacancy. The esthetic images indicate a sharp isotropic tip in this experiment.

We intentionally choose a clean area to study the intrinsic electronic properties of FeSe, as shown in Fig. 2a. A drift-correction algorithm is applied to remove the influence of thermal drift in the scanning process[6,7]. Fourier transform of the drift-corrected STM image is shown in Fig. 2b. Four pairs of Bragg peaks are separated into two groups with labels of ($Q_x$, $Q_y$) and ($Q_a$, $Q_b$). With $Q_x = (2\pi/a_0, 0)$ and $Q_y = (0, 2\pi/a_0)$, the group of ($Q_x$, $Q_y$) corresponds to the Se lattice. With $Q_a = (2\pi/a_0, -2\pi/a_0)$ and $Q_b = (2\pi/a_0, 2\pi/a_0)$, the group of ($Q_a$, $Q_b$) corresponds to the Fe lattice. The detection of ($Q_a$, $Q_b$) is related to the underlying electronic contribution from Fe atoms, although they are not visible in the topographic image. The very sharp Bragg peaks indicate a perfect periodic lattice in the drift-corrected topography[6,7], on top of which locations of Se atoms can be precisely determined. Locations of Fe atoms can be further determined from the middle point of two nearest Se atoms. The corresponding atomic locations are superimposed on the topography, as shown in a partial image in the inset of Fig. 2c.

To investigate spatial-dependent electronic properties, we measure a dense array (up to 17 pixels per unit cell) of $dI/dV$ spectrum as a function of the position $r$ and the variable voltage $V$, in the same area as in Fig. 2a. The pixel density is comparable with that in a similar analysis of the cuprate superconductor[8]. The $dI/dV$ spectrum at a specified location is proportional to the LDOS. Differential conductance maps $g(r, E) = dI/dV(r, E)$ are simultaneously obtained as a function of energy $E = eV$. With atomic positions determined, the differential conductance at positions of Fe$_1$ or Fe$_2$ atoms can be selectively extracted and averaged. The difference between the averaged differential conductance at sites of Fe$_1$ and Fe$_2$ atoms are calculated as a function of energy ($E = eV$). As shown in Fig. 2c, this difference gradually decreases with the increase of the energy, and approaches zero around the Fermi energy (zero bias). As a comparison, we also separate all the Fe atoms into two random groups and average the differential conductance in each group (see Supplementary Fig. 4). As shown in Fig. 2c, the difference between the averaged differential conductance at Fe sites in two random groups is almost zero, indicating that the difference at Fe$_1$ and Fe$_2$ sites is a real physical signal instead of fluctuations. In the cuprate superconductor, the similar definition has been

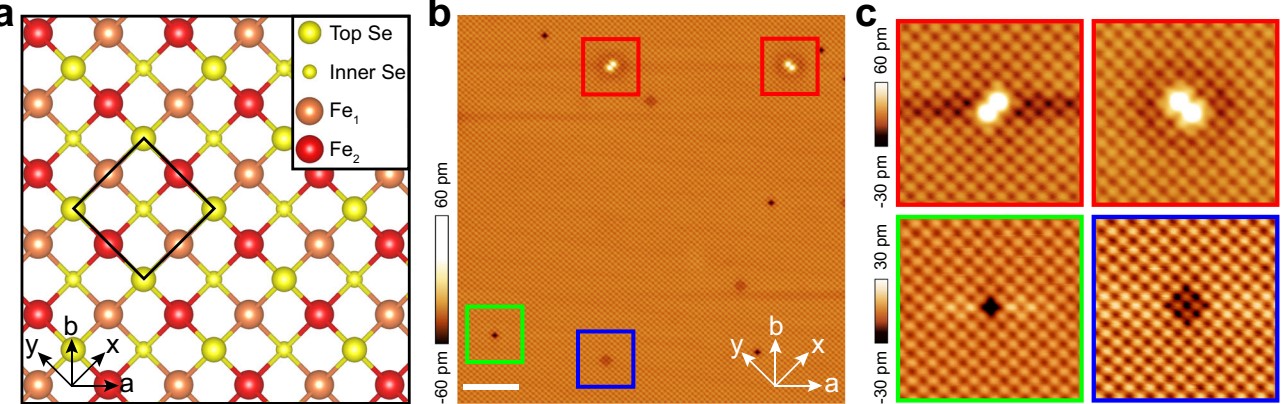

**Fig. 1 Crystal structure and surface characterization of FeSe. a** Top view of the schematic structure of FeSe. The $a$ and $b$ axes are defined by the Fe lattice. The $x$ and $y$ axes are defined by the Se lattice. The black diamond indicates the top layer Se lattice. Two different Fe sites are defined by their different bonding directions with the top layer Se atoms. **b** Atomic-resolved topographic image (35 nm × 35 nm) taken under $V_b = 100$ mV and $I_s = 20$ pA (scale bar: 5 nm). **c** The enlargement of the red, green, and blue squares (5 nm × 5 nm) in **b**. The tunneling condition is $V_b = 100$ mV and $I_s = 20$ pA.

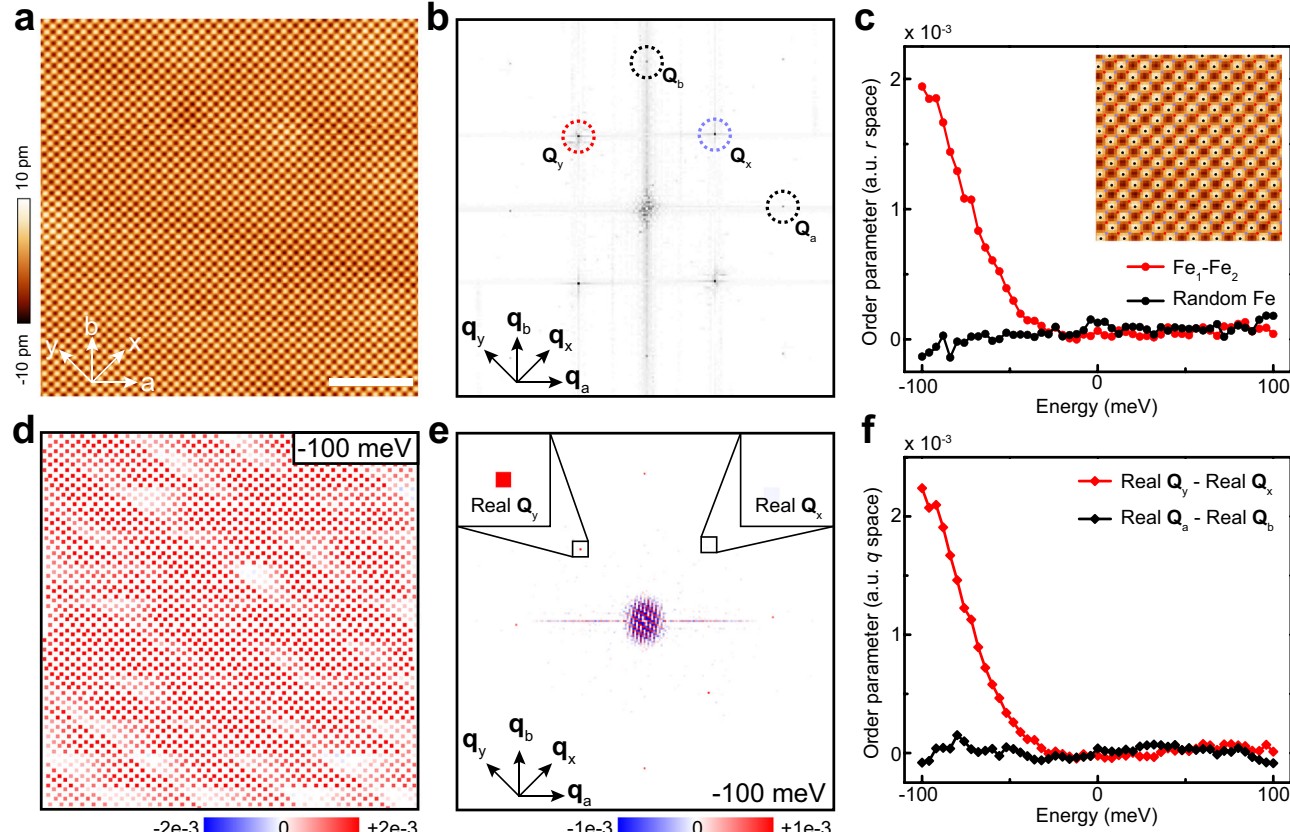

**Fig. 2 Observation of a [110] electronic order at liquid nitrogen temperature. a** An 18 nm × 18 nm STM image of a clean area under $V_b = 100$ mV and $I_s = 100$ pA (scale bar: 4 nm). **b** Fourier transform of the drift-corrected topography in **a**. Bragg peaks of the Se lattice ($\mathbf{Q}_x$, $\mathbf{Q}_y$) and Fe lattice ($\mathbf{Q}_a$, $\mathbf{Q}_b$) are labeled correspondingly. **c** The red solid dots are the difference between the averaged differential conductance at $Fe_1$ and $Fe_2$ sites in selected area. The black solid dots are the difference between the averaged differential conductance at two groups of random Fe sites. Inset shows a part of the drift-corrected topography with precise atomic positions superimposed on it. **d** The pattern of local [110] electronic order at −100 meV, with each dot representing the difference between differential conductance at $Fe_1$ and $Fe_2$ within one unit cell (scale bar: 2 nm). **e** Real part of the Fourier transformed conductance map at −100 meV. Insets show the enlargement of Bragg peaks. The intensities of the Bragg peaks are Re[$g(\mathbf{Q}_x, -100$ meV$)$] = 4.7 × 10⁻⁵ and Re[$g(\mathbf{Q}_y, -100$ meV$)$] = 2.3 × 10⁻³, respectively. **f** The red curve is the difference between real part of $\mathbf{Q}_x$ and $\mathbf{Q}_y$ (Re[$g(\mathbf{q} = \mathbf{Q}_y, E)$] − Re[$g(\mathbf{q} = \mathbf{Q}_x, E)$]). The black curve is the difference between real part of $\mathbf{Q}_a$ and $\mathbf{Q}_b$ (Re[$g(\mathbf{q} = \mathbf{Q}_a, E)$] − Re[$g(\mathbf{q} = \mathbf{Q}_b, E)$]).

named as a collective order parameter ($r$ space)[8]. An inequivalent differential conductance between two types of oxygen atoms leads to a nematic order along the Cu–Cu lattice direction. Due to the different structure between FeSe and CuO plane of the cuprates,

here the difference between the differential conductance at $Fe_1$ and $Fe_2$ sites represents a checkerboard charge order along the [110] direction of iron lattice. Normally an order parameter does not evolve with energy, as a traditional order described by the

Landau-order-parameter. The energy dependence of the order in STM experiment is related to an energy-dependent response of $dI/dV$ spectrum to the order after the order is somehow coupled in the spectrum.

The difference between the averaged differential conductance at $Fe_1$ and $Fe_2$ sites represents the collective order in the entire area. We further explore the strength of the local order and their spatial distribution, by calculating the difference between the differential conductance at $Fe_1$ and $Fe_2$ sites in each unit cell. To remove the influence of background noise, energy-dependent conductance maps have been slightly filtered before the calculation (see ref. [8] and Supplementary Fig. 5). In Fig. 2d, we show the local order within each unit cell at $-100$ meV. The local order shows values with the same sign (red color) in the entire area, representing a robust order. In contrast, the local order at 100 meV shows very weak signals and aggregated patches of different signs (Supplementary Fig. 5), leading to an averaged zero value of the collective order at 100 meV. The evolution of the local order is compatible with the energy-dependent collective order in Fig. 2c.

The order parameter can also be deduced from the $q$ space information. Along lines of $Fe_1$ or $Fe_2$ atoms, the electronic structure of Fe atoms exhibits the same periodicity as the Se lattice. For a conductance map at $-100$ meV, its Fourier transform $g(\mathbf{q}, E)$ is shown in Fig. 2e. We emphasize that the center of conductance map has been shifted to a site of Se atom before the Fourier transform. Then electronic information of Fe atoms can be extracted from the real part of Bragg peaks of $\mathbf{Q}_x$ and $\mathbf{Q}_y$[6,7]. As shown in the insets of Fig. 2e, $Re[g(\mathbf{q} = \mathbf{Q}_x, E)]$ and $Re[g(\mathbf{q} = \mathbf{Q}_y, E)]$ show obvious inequivalent intensities, despite of their small values. We define the difference between $Re[g(\mathbf{q} = \mathbf{Q}_y, E)]$ and $Re[g(\mathbf{q} = \mathbf{Q}_x, E)]$ as a collective order ($q$ space), similar to the definition in cuprate superconductor[6,8]. As shown in Fig. 2f, the energy dependence of this order is quantitatively consistent with that in Fig. 2c. The checkerboard charge order is thus confirmed in both $r$ space and $q$ space at 77 K. We have tested different samples with different tips under different bias voltages, and this order is a robust phenomenon (Supplementary Fig. 6).

In most previous reports about FeSe, electronic nematicity is found to be along the [100] direction of Fe lattice, namely, the [100] nematic order. In previous STM experiments, the intrinsic nematic order is pinned by dumbbell defects. Electronic nematicity is mainly determined by unidirectional depressions straddling dumbbell defects. Here at 77 K, the unidirectional depression is very weak and cannot be observed around dumbbell defects under our tunneling condition (Fig. 1b). We have also tried to extract the [100] nematic order from $q$ space. For the clean area we choose, we check the intrinsic [100] nematic order in $q$ space by comparing the intensity of $Re[g(\mathbf{q} = \mathbf{Q}_a, E)]$ and $Re[g(\mathbf{q} = \mathbf{Q}_b, E)]$. With the periodic electronic structure of Fe atoms along $a$ and $b$ axes considered, the center of conductance map is shifted to a site of Fe atom before the Fourier transform. The difference between $Re[g(\mathbf{q} = \mathbf{Q}_a, E)]$ and $Re[g(\mathbf{q} = \mathbf{Q}_b, E)]$ is quantitatively small (see the black curve in Fig. 2f). We thus conclude a rather weak [100] nematicity at liquid nitrogen temperature, at least under this tunneling condition.

**Intrinsic electronic orders at 4.5 K**. We further study the intrinsic ordered states at a much lower temperature at $T = 4.5$ K. To be consistent with the previous reports[43,57], we define the longer axis of the Fe lattice as the nominal $a$ axis. A twin boundary (TB) appears in the asgrown single crystal of FeSe, and the number of TB increases with the decrease of temperature. On two sides of each TB, the $a$ and $b$ axes are rotated by 90°. Figure 3a shows an atomic-resolved large area STM image at 4.5 K. There

are two TBs in this field of view (FOV), as shown by the wide dark stripes in Fig. 3a. A unidirectional stripe straddles each dumbbell defect, with the direction rotated by 45° from the Se lattice. On one side of the TB, the unidirectional stripes of different dumbbell defects are all along the same direction regardless of the dumbbell direction. On another side of the TB, the unidirectional stripes are rotated by 90°. This unidirectional stripe at a high bias voltage ($V_b = 100$ mV) is a correspondence of the unidirectional depression at low bias voltage[57]. For the same dumbbell defect, the unidirectional pattern under the high bias voltage is rotated by 90° compared to that under the low bias voltage. This could be because that the unidirectional patterns are originated from the scattering between the $d_{yz}$ orbitals of Fe atoms[43]. Consistent with previous STM reports of FeSe[37,39,43,57], this unidirectional feature is a direct evidence of the [100] nematicity.

To study the intrinsic ordered states at 4.5 K, we select two clean areas on two sides of one TB, as shown by the black and blue squares in Fig. 3a. The zoom-in images are measured with a relatively low bias voltage $V_b = 20$ mV and shown in Fig. 3b, e, respectively. We intentionally choose this bias voltage to reduce the influence of setup effect on the [110] electronic order (see Supplementary Note 5). On each side of the TB, a rough square lattice shows a similar lattice constant along $a$ and $b$ axes. For both selected topographies $T(r)$, we do the drift-correction, shift the center to a site of Fe atom, and calculate the Fourier transform. The results are shown in Fig. 3c, f, respectively. In Fig. 3c, there is a clear stronger intensity of $Re[T(\mathbf{q} = \mathbf{Q}_a)]$ compared with that of $Re[T(\mathbf{q} = \mathbf{Q}_b)]$, indicating a [100] nematicity. With $a$ axis rotated by 90° on the other side of the TB, the peaks with stronger intensity at $\mathbf{q} = \mathbf{Q}_a$ are also rotated by 90°, as shown in Fig. 3f. This further confirms the intrinsic [100] nematicity in a clean area.

We further measure a dense array (10 pixels per 1 nm) of $dI/dV$ spectra in the same FOV as in Fig. 3b, e. For the series of conductance maps $g(r, E)$, we shift the map center to a site of Fe atom, calculate the Fourier transform, and extract $Re[g(\mathbf{q} = \mathbf{Q}_a, E)]$ and $Re[g(\mathbf{q} = \mathbf{Q}_b, E)]$ correspondingly. The difference between $Re[g(\mathbf{q} = \mathbf{Q}_a, E)]$ and $Re[g(\mathbf{q} = \mathbf{Q}_b, E)]$ are shown as a function of $E$ in Fig. 3d, g, for a clean area on two sides of the TB. On both sides of the TB, there is a clear trend of nematic order for both positive and negative energies. The order increases with increasing $|E|$ and the order at negative energy is about three times stronger. The trend of nematicity is nearly opposite when across the TB. A small quantitative difference is observed, which may be due to some inhomogeneity of the nematicity.

We also investigate the [110] electronic order at 4.5 K. To extract the local order within each unit cell, conductance maps are filtered similar to that at 77 K to remove the influence of background noise. Applying the same method as that at 77 K, we extract the differential conductance of each $Fe_1$ and $Fe_2$ atoms from the filtered conductance map. Afterward the difference between the differential conductance at $Fe_1$ and $Fe_2$ sites in each unit cell is calculated. At a representative energy of $-100$ meV, the local order is shown in Fig. 4a, c, at two sides of the TB. For the right side of the TB, the local order is dominated by the negative order (blue dots, Fig. 4c). For the left side of the TB, a domain wall is roughly along the diagonal direction, dividing the region into two parts with different signs of local order (Fig. 4a). The upper-left corner of this FOV is dominated by the positive order (red dots), and the lower-right corner of this FOV is dominated by the negative order (blue dots). The simultaneous observation of different signs of local order excludes the possibility that the observed charge order is directly from two chemically inequivalent Fe sites.

We further average the collective [110] electronic order on two sides of the TB. The collective order is calculated by averaging the

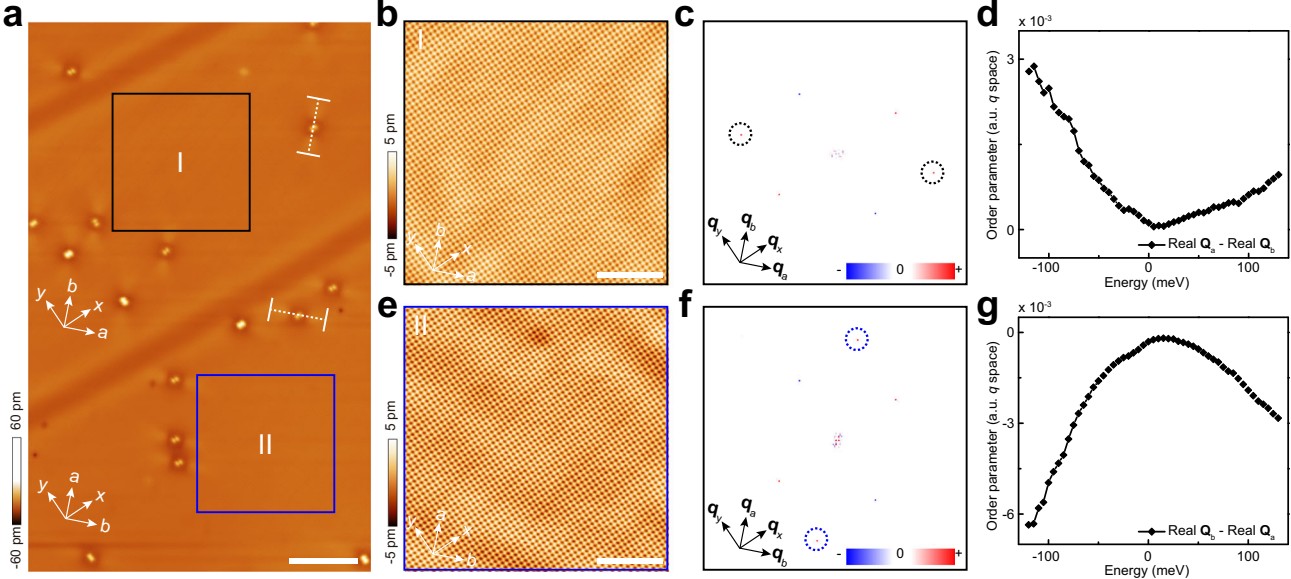

**Fig. 3 Nematicity along [100] direction measured at liquid helium temperature. a** A large area STM image with two twin boundaries (50 nm × 80 nm, $V_b$ = 100 mV, $I_s$ = 20 pA, scale bar: 10 nm). The $a$ and $b$ axes are rotated by 90° when across each twin boundary. **b** The enlargement of the black square (20 nm × 20 nm, $V_b$ = 20 mV, $I_s$ = 20 pA) in **a** (scale bar: 5 nm). **c** Real part of Fourier transform of the drift-corrected topography in **b**. The black dashed circles highlight the stronger Bragg peaks of the Fe lattice along $Q_a$ axis. **d** Difference between Re[$g(\mathbf{q} = \mathbf{Q}_a, E)$] and Re[$g(\mathbf{q} = \mathbf{Q}_b, E)$] versus the energy. **e**–**g** The same as that in **b**–**d** but with the area on the other side of the twin boundary, as indicated by the blue square in **a**. The size is also 20 nm × 20 nm with setup condition $V_b$ = 20 mV and $I_s$ = 20 pA. Scale bar in **e** is 5 nm. The tunneling conditions for the grid spectroscopy measurements are the same as that of the corresponding topographies.

local order in the selected area. For the right side of the TB, the collective order is shown in Fig. 4d. For the left side of the TB, the collective order over the FOV of Fig. 4a is quantitatively small (see the black curve in Fig. 4b). The rather small collective order is due to the two opposite domains. The appearance of nearby domains with opposite signs of orders exclude that the observed order is caused by the tip anisotropy[7]. The red and blue domains evolve smoothly versus energy (Supplementary Fig. 7). We thus calculate the collective order of two domains separately. The results are shown in Fig. 4b for the red and blue domains, respectively. From Fig. 4b, d, a similar trend is found for the [110] electronic order. Different from the [110] order at 77 K, there is a gap feature with an almost zero order in the energy range around [−40, 40] meV. Outside the gap, the order increases with increasing |E| for both positive and negative energies, and the order at negative energy is about three times stronger. The clearer gap in Fig. 4d is due to the averaging of stronger signals in this domain. We have tested different samples, showing a similar trend of the [110] electronic order, as well as the domain structure (Supplementary Fig. 8). With $a/b$ axes spontaneously fixed between two TBs, it is reasonable to observe domains in the [110] electronic order. We also measure an area that is far from the TB at 4.5 K (Supplementary Fig. 9). A robust [110] electronic order and a clear gap are both observed. The domain structure means that the correlation length of the [110] electronic order at 4.5 K is shorter than that at 77 K. From the robust order far from the TB, we presume that the [110] electronic order with domains are possibly induced by the ubiquitous twin boundaries at low temperatures. We also emphasize that both the [100] nematic order and the [110] electronic order are reproducibly detected with the slow STM measurement, thus can be characterized as static orders.

## Discussion

The intrinsic [100] nematic order is mainly detected at 4.5 K, consistent with previous STM experiments in which the order is

determined from unidirectional signals around dumbbell defects[57]. The main discovery of this work is the observation of an intrinsic [110] electronic order (checkerboard charge order) at both 77 and 4.5 K. The [110] electronic order is much stronger than the [100] nematicity at 77 K. At 4.5 K, the [100] nematicity is enhanced compared with that at 77 K. The local [110] electronic order is robust at 77 K, while domains appear at 4.5 K. Phenomenologically, we presume a competition between these two electronic orders.

The observation of two electronic orders along different directions reminds of the INS results of bulk FeSe, in which both the stripe and Néel spin fluctuations are detected at ($\pi$,0) and ($\pi$, $\pi$), respectively[51]. Both the direction and wavelength are the same for the Néel spin fluctuations and the [110] electronic order. The strength of the Néel spin fluctuations decrease with decreasing temperature, while the strength of the stripe spin fluctuations increase with decreasing temperature. The competition between two electronic orders is similar to the competition between two spin fluctuations. Furthermore, the gap feature in the [110] electronic order is consistent with the gap of Néel spin fluctuations at liquid helium temperature. The discrepancy is that although spin fluctuations can be coupled in the d$I$/d$V$ spectra as an inelastic tunneling phenomenon[62–64], we cannot straightforwardly explain the [110] local order pattern with inelastic tunneling spectroscopy. Although there is a possibility that the Néel spin fluctuations and the [110] electronic order are two unrelated phenomena, this possibility would be relatively very small based on the above simultaneous similarities.

A non-magnetic tip is used in our STM experiment, then the measured LDOS is related to the electronic structure in the charge channel. Some previous examples present how the non-magnetic tip measures signals in the spin channel. Lawler et al. have detected a small difference of differential conductance at $O_x$ and $O_y$ sites within each unit cell in $Bi_2Sr_2CaCu_2O_{8+\delta}$. The intra-unit cell symmetry breaking is expected to be connected with the static Néel magnetic order[6]. In the epitaxy film of stoichiometric FeTe,

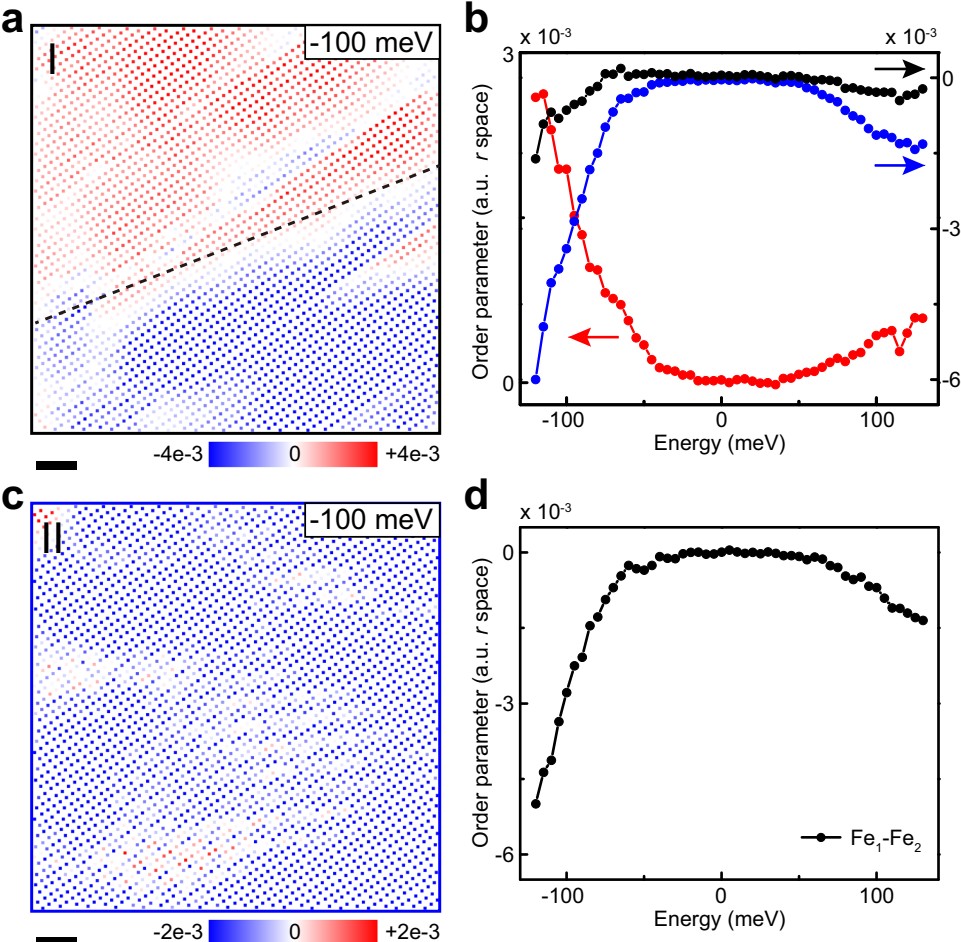

**Fig. 4 [110] Electronic order measured at liquid helium temperature. a** Local [110] electronic order on the left side of the twin boundary (scale bar: 2 nm). The black dashed line highlights the boundary of the domains with different signs. **b** The red and blue curves are collective orders for the red and blue areas in **a**, respectively. The black curve is the collective order over the area of **a**. **c** Local [110] electronic order on the right side of the twin boundary (scale bar: 2 nm). **d** The collective order on the right side of the twin boundary.

a claimed charge order has been revealed although the periodicity is the same as the underlying spin order[65]. In both examples, the wavevector of the detected signal is the same as that of the spin order. The signal in the spin channel may be weakly coupled in the charge channel, while the weak signal is detectable in delicate STM measurements. In our experiment, the wavevector of the checkerboard charge order along [110] direction is the same as that of a Néel order.

On the other hand, the [100] nematic order is manifested by the 90° rotation of the stronger Fe Bragg peaks when across the TB, compatible with unidirectional signals around dumbbell defects[37]. This static [100] nematic order detected in our experiment is deduced from two inequivalent Fe Bragg peaks with wave vectors of $(2\pi,0)$ and $(0,2\pi)$. Here we unify the basic vector to be the single iron lattice in real space. The wave vector of the stripe spin fluctuations should be $(\pi,0)$ based on this notation. The twice wave vector of the [100] nematic order compared with the stripe spin fluctuations is compatible with an accompanying charge order whose wavelength is half of that of the spin-order[66,67].

An alternative explanation for the wave vector of [100] nematic order may originate from the orbital redistribution of each Fe atom. The splitting of the $3d_{xz}$ and $3d_{yz}$ orbital of FeSe in orthorhombic phase has been extensively studied by ARPES[30–35]. Further theoretical calculations are necessary to clarify whether the different electronic state of $Fe_1$ and $Fe_2$ atoms is related with the orbital effect.

It is desirable for the [110] electronic order to be confirmed with other different experimental techniques. Some characteristic features should be brought to researchers' attention. The energy scale in our experiment is extended to ±100 meV for the detection of this electronic order. If a technical tool focuses on a much smaller energy scale, this order cannot be detected. In addition, the [110] electronic order forms nanometer-sized domains at low temperatures, which requires a local probe with atomic spatial resolution. A detwined sample is also worth to be explored[36]. The signal of electronic orders in our experiment is very small (but reproducible). The rather small signal requires a precision measurement to distinguish two inequivalent Fe sites and detect the checkerboard charge order.

In conclusion, we investigate the intrinsic ordered states of FeSe at 77 and 4.5 K with STM. An exotic [110] electronic order is observed together with the traditional [100] nematic order. The [110] electronic order is robust within each unit-cell at 77 K. A domain pattern appears at 4.5 K, leading to a shorter correlation length of the [110] electronic order. In contrast, the [100] nematicity is enhanced with the decrease of temperature. The evolution of these two orders share similar behavior with the Néel and stripe spin fluctuations. Recently, a calculation based on the group theory has predicted a similar charge order in iron-based superconductors[68]. Our work lays a foundation for further experiments and theories to probe the nature of various ordered states in FeSe.

## Methods

**Sample preparation**. FeSe single crystals were grown under a permanent gradient temperature of ~400–330 °C in the KCl–AlCl$_3$ flux. After one month of growth, large patches of FeSe flakes with a typical size of 2 mm × 2 mm were obtained. The X-ray diffraction (XRD) measurement shows that it is stoichiometric without other impurity phases[51]. The transport measurement shows a sharp resistivity drop at 8.9 K and the temperature transition width is measured to be 1.7 K. The residual resistivity ratio (RRR, defined as $R_{300K}/R_{11K}$) is ~20, indicating the high quality of the samples (see Supplementary Fig. 1).

**STM measurement**. STM and STS experiments were carried out in a commercial system with ultra-high vacuum and low temperatures. The samples were pasted on a beryllium copper plate by an EPOTEK silver glue and were then fixed on the sample holder. The single crystals were cleaved at liquid nitrogen temperature and inserted into the STM head immediately. A tungsten tip was prepared by the electrochemical etching method. It was then dealt with electron beam sputtering and field emission on a single crystalline of Au (111) surface. The optimization of tip condition for this experiment is presented in Supplementary Fig. 2. The base vacuum was better than $5 \times 10^{-11}$ Torr. The STM images were taken under a constant current mode with a feedback loop control. The d$I$/d$V$ spectra were taken at a bias modulation of 2 mV with a modulation frequency of 1213.7 Hz.

## Data availability

All the data related to this study are available upon reasonable request from the corresponding author.

## Code availability

The original codes used in this study are available upon reasonable request from the corresponding author.

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

## Acknowledgements

This work was supported by the National Key R&D Program of China (Grant No. 2019YFA0308602), the Key R&D Program of Zhejiang Province, China (2021C01002), the National Natural Science Foundation of China (NSFC-11374260), and the Fundamental Research Funds for the Central Universities in China. The work at Fudan University was supported by the Innovation Program of Shanghai Municipal Education Commission (Grants No. 2017-01-07-00-07-E00018), the National Natural Science Foundation of China (NSFC-11874119), and the National Key R&D Program of China (Grant No. 2016YFA0300203).

## Author contributions

K.L.B., W.H.Z., and Y.F. conducted the STM experiment. Z.X.W. participated in the experiment. Y.Z. and F.Z.A. discussed the results. Q.S.W., H.L.W., and J.Z. grew the samples. K.L.B. analyzed the data. Y.Y. and K.L.B. wrote the paper. Y.Y. supervised the experiment. All authors have read the paper and approved it.

## Competing interests

The authors declare no competing interests.
