## [Peer Review File · Nature Communications]

Reviewers' comments:

Reviewer #1 (Remarks to the Author):

The manuscript by Bu and coworkers reports observation of nematicity in the [110] direction in FeSe, which is in addition to the well-known [100] nematicity. They find the nematicity through detailed analysis of low temperature STM data. The STM data is of high quality and the authors present a sophisticated analysis. The analysis reveals tiny differences in the differential conductance between the Fe1 and Fe2 positions, which the authors interpret as due to the [110] nematicity. The manuscript is well written and suitable for publication in a journal such as Nature Communications.

Given that the differences in the differential conductance are so tiny, one wonders how robust this effect is, and it may be worth for the authors to provide some additional analysis (as detailed below) to underpin their argument. The reader is left with a bit of doubt how much of the effect is real, and how much could be, e.g., numerical errors of the algorithm.

With such additional analysis, I expect that the manuscript is suitable for publication.

* Please add color bars with scale to topographic images

* B1g and B2g orders should be defined in fig. 1a, also please make clear which coordinate system the notation [100] and [110] refers to (a/b or x/y)

* What is the y axis of fig. 2c? Is this nS, or a.u.? Please provide units. What means “random Fe”? It would be useful to show for comparison Fe1+Fe2

* Is it worth defining a normalized order parameter, such as $(g_{Fe1}-g_{Fe2})/(g_{Fe1}+g_{Fe2})$? This would be unit-less and thus free of the setup condition

* Pg. 4 “... we measure a dense array ...” – please define what is meant by a dense array. How sure can the authors be that discretization/interpolation does not play a role in the calculation of the order parameter?

* it is slightly worrying that the images need to be filtered on a length scale of 1nm before the analysis is carried out - this suggests that the filtering averages or smoothens over a longer length scale than the features that are investigated, and hence suppresses them. are the results robust against the choice of the filter radius? Do maps taken in the same area with different parameters (e.g. density of pixels) yield the same result?

* does the bias dependence of the nematicity (as shown, e.g., in fig. 4 b,d) depend on the setpoint condition? I.e. is it the same whether the setpoint is +100mV or -100mV?

* does analysis of maps taken with different density of points in the same area yield the same results on the [110] nematicity?

Reviewer #2 (Remarks to the Author):

The authors observed two different nematic orders in an Fe-based superconductor FeSe using scanning tunneling microscopy and spectroscopy at 77 K and 4.5 K. The anisotropic electronic states

have been widely observed in the various unconventional superconductors. In general, the nematicities (B_{1g}) accompanied with collinear AFM spin order are developed along the [100] direction in Fe-based superconductors. Here, they observed the long-ranged B_{2g} nematicity along the [110] at 77 K. The coherence length of B_{2g} is smaller while the B_{1g} is enhanced at 4.5 K.

As the authors pointed out, the nematicity in unconventional superconductors has been intensively investigated with various approaches. Nonetheless, we haven't fully understood the origin of the nematicity and its relation with superconductivity. The coexistence of the B_{1g} nematicity and the collinear AFM spin order have been widely observed in the Fe-based superconductors. In the case of absence of spin ordering, the nematicity along different direction has been recently observed in the heavily doped Fe-base superconductors. Therefore, authors consider that the paramagnetic bulk FeSe is one of the candidates that has different nematicity from B_{1g} and provide a good platform to study the origin of the nematicity.

Compared with importance of their message, the data and analysis seems not sufficient to back up the strong claims made in their discussion. Unless the authors can appropriately rebut the comments below, my evaluation would be that the results, while interesting, are insufficient for publication in Nature Communications.

First, the authors visualize the B_{2g} nematicity along the [110] direction in the surface of FeSe sample using spatially resolved STS at 77 K (Fig. 2). It is different from the B_{1g} nematicity along the [100] direction that has been observed in this material. They used two different approaches to quantify the B_{2g} nematicity. One is acquiring the difference of the spectra of two different Fe sites and another comparing two FFT spectral intensities at the reciprocal lattice points corresponding to Se lattice. They seem to show strong anisotropic features along the [110] direction. In order to be more convincing to the reader, the authors should

- 1) Provide evidence that their tip is sufficiently isotropic.
- 2) Show the acquired dI/dV spectra on Fe1, Fe2, Se atoms.
- 3) Present data acquired with $V_B = -100 \text{ mV} < 0$ and $I_{\text{set}} = 100 \text{ nA}$ or reasonable explanations to exclude the set-up effect. The larger variations in the negative bias seem to originate with the positive bias set-up.
- 4) In addition, I can see the impurities (weakly darker and brighter contrast) in Fig. 2a and the suppressed nematic order in Fig.2b. even though they intentionally choose the field of view without defects to investigate intrinsic electronic properties. Do they have any correlation between impurity features in the STM images and local suppression in Fig. 2b? What is the origin of local suppression of B_{2g} nematicity?

Second, the authors quantify the B_{1g} nematicity as a function of bias using spatially resolved STS at 4.5 K (Fig. 3). The unidirectional feature close to the defect as an evidence for B_{1g} nematicity is consistent with previous STM studies on FeSe. STM images of two different domains and their

anisotropic intensities of their FFT spectra also support the existence of the B_{1g} nematic order. They also did similar analysis for dI/dV maps as a function of bias. The higher bias show the stronger nematicity. In order to make their statements more concrete, the authors should

- 5) Compare the acquired dI/dV spectra with previous STS study of this material.
- 6) Put the set-up conditions of dI/dV measurements and compare results acquired with two different polarities of set-up bias?
- 7) Explain why the higher bias voltages show the stronger nematicity and the nematicity is more clear at the negative bias.
- 8) Put the length scale of the dashed lines in Fig. 3a and explain what they represent.
- 9) Did they normalize the FFT spectral intensities corresponding to B_{1g} nematicity by using the reference value (for example, FFT spectral intensities for Se lattice) to plot order parameter as a function of bias voltage?

Third, the authors quantify the B_{2g} nematicity as a function of bias using spatially resolved STS at 4.5 K (Fig. 4). Contrast to results at 77 K, the negative and positive nematicity coexist and their strength is not zero at positive bias. The order parameter as a function of bias show the gap-like feature whose size is comparable to the gap of Neel spin fluctuation. In this section, the authors should

- 10) Show the acquired dI/dV spectra on Fe1, Fe2, Se atoms for positive, negative, zero order parameter region, respectively. Compare their spectra with ones acquired at 77 K.
- 11) Check the possibility that the asymmetric feature in the order parameter spectra come from the set-up effect. If they have data acquired with opposite polarity set-up bias, compare the results with presented one in the manuscript.
- 12) Explain the origin of energy dependency of domain structure shown in Fig. S5?
- 13) Explain why the negative order parameter seems to have larger and clear gap than the positive one in Fig. 4b?

Reviewer #3 (Remarks to the Author):

The others report an STM study of clean FeSe single crystals at liquid nitrogen (77K) and liquid He (4.5 K) temperatures. Their main novel finding is a difference in the differential conductance between “Fe1” and “Fe2” sites, where the Fe1 and Fe2 sites are arranged in an alternating fashion.

This difference has been dubbed “real space B2g nematic order” and its value is strongly energy dependent and also temperature dependent.

The data appear to be of high quality and the manuscript is well organized and – for the most part – clearly written. However, many questions remain open.

The so-called Fe1 and Fe2 sites are distributed in a checkerboard like manner. In consequence, a difference between the averaged differential conductance at sites of Fe1 and Fe2 atoms does, as far as I can see, not imply a *nematic* order, which is defined as a rotational symmetry breaking. The authors write “Here we note that each type of Fe atoms are rotated by 45° with the [100] direction of the Fe lattice (Fig. 1a). The difference between the differential-conductance at Fe1 and Fe2 sites thus represents an electronic anisotropy along the [110] direction, which has a form of B2g symmetry.” This reasoning is unclear to me. Indeed, the authors discuss how this exotic order might couple to Neel-type magnetic fluctuations (which also are not nematic). What makes the authors choose the term “B2g nematic” for this real-space order?

In the bulk crystals, “Fe1” and “Fe2” sites are symmetry equivalent. What is the role of the surface in breaking the symmetry between the “top Se” and “Inner Se” atoms? May this have an effect on the observed unusual symmetry breaking?

What are the implications of an energy-dependent order parameter of a static order? Do the authors have a traditional Landau-order-parameter framework in mind, as is often the case in the field?

The dI/dV spectra are of primary importance for the energy dependence of the effect. The authors should show a representative selection of them, at least in the supplement.

In summary, the data seem very nice and potentially also very interesting. However, these open questions make it difficult for me to recommend publication.

Reply to reviewers:

Our replies are in blue color and the reviewers' comments are in black color.

Reviewer #1 (Remarks to the Author):

The manuscript by Bu and coworkers reports observation of nematicity in the [110] direction in FeSe, which is in addition to the well-known [100] nematicity. They find the nematicity through detailed analysis of low temperature STM data. The STM data is of high quality and the authors present a sophisticated analysis. The analysis reveals tiny differences in the differential conductance between the Fe1 and Fe2 positions, which the authors interpret as due to the [110] nematicity. The manuscript is well written and suitable for publication in a journal such as Nature Communications.

Given that the differences in the differential conductance are so tiny, one wonders how robust this effect is, and it may be worth for the authors to provide some additional analysis (as detailed below) to underpin their argument. The reader is left with a bit of doubt how much of the effect is real, and how much could be, e.g., numerical errors of the algorithm.

With such additional analysis, I expect that the manuscript is suitable for publication.

We totally agree with the reviewer that the detected signal is very small. We have been very careful to check the data and analysis to make sure they are real physical signal, instead of numerical errors from the analysis. In the revised manuscript, we supplied more datasets both at 77 K and at 4.5 K. The [110] nematic order is robust against the setup condition and the density of pixels. The consistency among different datasets taken by different tips on different samples strengthens our claims about the [110] nematicity.

* Please add color bars with scale to topographic images

In the revised manuscript, we added color bars with scale to all topographic images.

* B1g and B2g orders should be defined in fig. 1a, also please make clear which coordinate system the notation [100] and [110] refers to (a/b or x/y)

We removed the terms of B_{1g} and B_{2g} to avoid potential confusions caused by different definitions. Instead, we only use the terms of [100] and [110] nematicity to describe the symmetry breaking along two different directions. In the revised manuscript, we added 'both the notation [100] and [110] are defined based on the a/b axis of nominal Fe lattice'. The [100] direction corresponds to the nearest Fe-Fe direction, while the [110] direction corresponds to the diagonal Fe-Fe direction.

* What is the y axis of fig. 2c? Is this nS, or a.u.? Please provide units. What means "random Fe"? It would be useful to show for comparison Fe1+Fe2

The units of the y axis in Figs. 2c, 2f, 3d, 3g, 4b and 4d are arbitrary units. In the revised manuscript, we amended the y axis labels of these figures.

After precisely locating all sites of Fe atoms, we randomly assign each Fe atom into one of two groups by using computer-created random numbers. The spectra of each group of random Fe atoms are averaged respectively. The data presented by black dots in Fig. 2c is the difference between the averaged spectra of two groups of random Fe atoms, which we define as the order parameter of 'random Fe'. In the revised manuscript, we showed the sites of these two groups of random Fe atoms

in Fig. S3c.

In Fig. S3b, we show the averaged spectra of Fe₁ and Fe₂ atoms, from which a weak but *non-zero* difference can be discerned. We emphasize that each average spectrum has been averaged for more than 2250 times for the dataset in an area of 18 nm × 18 nm (Fig. 2 and Fig. S3). An analysis for a larger area would involve more counting times for the averaged spectra. The key of the drift-correction algorithm is to precisely align atomic sites to enable averaging spatially-located spectrum, leading to a result with much-enhanced signal to noise ratio. With this much enhanced signal to noise ratio, the tiny difference between two averaged spectra could be reproducibly discerned (inset of Fig. S3b). As a comparison, the difference between the averaged spectra of two groups of random Fe atoms is almost zero (inset of Fig. S3d).

* Is it worth defining a normalized order parameter, such as $(g_{\text{Fe1}}-g_{\text{Fe2}})/(g_{\text{Fe1}}+g_{\text{Fe2}})$? This would be unit-less and thus free of the setup condition.

We understand that the setup effect could be eliminated in this normalization. However, we do not make further analysis with this procedure because of following reasons. (1) The difference between two averaged dI/dV spectra is a weak signal. Although a complex data analysis has been made, we tried to avoid further data analysis if possible. More steps will introduce further analysis errors. (2) Due to the near-zero value of dI/dV near the Fermi level, using the normalized order parameter $(g_{\text{Fe1}}-g_{\text{Fe2}})/(g_{\text{Fe1}}+g_{\text{Fe2}})$ would bring large numerical errors.

For the referee's concern about the setup condition, a comprehensive discussion is provided later in this reply.

* Pg. 4 "... we measure a dense array ..." – please define what is meant by a dense array. How sure can the authors be that discretization/interpolation does not play a role in the calculation of the order parameter?

For a line of periodic cosine function, at least 3 pixels are needed to extract the main feature. Then at least $(2n+1)$ points are needed to depict a cosine oscillation with total n periodicities. In Fig. 2a, the size of the topography is 18 nm × 18 nm. With a lattice constant of FeSe $a_0 \approx 0.37$ nm, at least 98×98 pixels are required to map out the main feature of lattice. In our experiment, we map the topography of Fig. 2a (as well as the simultaneous grid spectroscopy) with a pixel density of 200×200 pixels, much larger than the required value. The pixel density in Figs. 3b and 3e are 200×200 pixels for an area of 20 nm × 20 nm, also much larger than the required minimum density. In the revised manuscript, we added a note to the 'dense array' as '... we measure a dense array (up to 17 pixels per unit cell) ...'.

The drift-correction (or Lawler-Fujita) algorithm used for our analysis was first developed in the research of cuprate superconductors [Nature 466, 347 (2010), Science 344, 612 (2014)]. With this algorithm, atomic sites can be precisely determined. The extracted dI/dV spectra are averaged for specific sites. The averaging procedure greatly enhances the signal to noise ratio, and a very weak signal can be discerned. This algorithm has been applied in various experiments to study the delicate electronic structure of different materials [Nat. Nanotechnol. 10, 849-853 (2015), Nat. Mater. 14, 318–324 (2015), Phys. Rev. X 5, 031022 (2015), PNAS 115, 6986-6990 (2018), etc.], proven to be a very powerful and reliable algorithm.

In the data analysis, the discretization/interpolation will truly introduce numerical errors to order parameters. With discretized sites normally not at precise atomic sites, data in nearest pixels have

to be used in the analysis. However, this error is pretty small when the pixel number is large enough. For all examples in previous literatures, the general practice is to choose a big enough pixel number, instead of discussing the pixel-number dependent errors. For the cuprate superconductor, a pixel density of 256×256 pixels has been used for analyzing the nematic order in an $27 \text{ nm} \times 27 \text{ nm}$ area [Sci. Rep. 7, 8059 (2017)]. With a similar lattice constant between FeSe and the cuprate superconductor, the density of pixels in our experiment is even a bit larger than that in cuprate experiments.

The pixel density in Figs.S5f, S5p and S5u are 200×200 pixels for an area of $20 \text{ nm} \times 20 \text{ nm}$. In Fig. S5a, the dataset is taken over a $12 \text{ nm} \times 12 \text{ nm}$ area with a larger pixel density of 160×160 . The trend of the [110] nematicity is robust against the density of pixels, which indicates that a pixel density of 10 pixels per 1 nm is enough for a reliable analysis.

* it is slightly worrying that the images need to be filtered on a length scale of 1nm before the analysis is carried out - this suggests that the filtering averages or smoothens over a longer length scale than the features that are investigated, and hence suppresses them. are the results robust against the choice of the filter radius? Do maps taken in the same area with different parameters (e.g. density of pixels) yield the same result?

In the original Lawler-Fujita algorithm [Nature 466, 347 (2010)], the nematic order is defined in the FFT-transformed dI/dV maps. Two perpendicular Bragg peaks (Q vector of the order) show different intensities, with the intensity difference representing the nematic order. This nematic order in the momentum space is related with the spectral difference at O_x and O_y sites in the real-space measurement. Then we call this order a *collective* order averaged in the scan area. The length scale of the collective order ($\sim 20 \text{ nm}$) is much larger than the intra-unit-cell scale ($\sim 0.5 \text{ nm}$).

When we try to calculate the intra-unit cell *local* order [Sci. Rep. 7, 8059 (2017)], we found a noisy and irregular pattern. It is puzzling why the nematic order is very clear in the momentum space but shows an irregular pattern in the real space. The answer is that, the Q vector peak in momentum space automatically filters out the noisy signal at different Q values, especially the long wavelength noises. Then we know how to obtain a clean real-space *local* order pattern, by filtering the FFT-transformed maps, making an inverse Fourier transformed real-space map, and then calculating the corresponding *local* order. For the cuprate superconductor, we have tested the filter-size dependent local-order patterns [Sci. Rep. 7, 8059 (2017)]. When the filtering effect is good enough, the local-order pattern will be independent of the filter size. Because the lattice constant of FeSe is similar to that of the cuprate superconductor, we empirically choose a filter size similar to that used in [Sci. Rep. 7, 8059 (2017)].

Physically, the Gaussian filter is applied to remove the influence of long-wavelength background noise. Then it is not a problem for the filter size to be larger than the intra-unit cell.

For comparison, we average the value of the local anisotropy for each energy we measured. Figure R1 shows the collective order without filtration and the averaged local anisotropy after the filtration. The data is extracted from the dataset in Fig. 2. As shown in Fig. R1, the collective order and averaged local anisotropy are quantitatively in agreement with each other.

Fig. R1 **a** Collective order reproduced from Fig. 2c in the main text. **b** Averaged local anisotropy after the Gaussian filtration.

* does the bias dependence of the nematicity (as shown, e.g., in fig. 4 b,d) depend on the setpoint condition? I.e. is it the same whether the setpoint is +100mV or -100mV?

Following we give a comprehensive discussion about the setup condition, which is one of our concerns in the detection of the small signal in this experiment. This discussion is also added in the revised manuscript.

A physical picture is that the nematic order is coupled in the dI/dV spectrum, from which we detect an energy-dependent *response* to the nematic order. Then we start from the measurement of a dI/dV spectrum.

The tunneling current in an STM experiment can be written as,

$$I_s(r, z, V_b) = f(r, z) \int_0^{eV_b} N(r, \varepsilon) d\varepsilon, \quad (1)$$

in which I_s is the setpoint current, r is the location of the tip in the ab plane, z is the distance between tip and sample, V_b is the bias voltage, and $N(r, \varepsilon)$ is the LDOS of the sample at position r and energy ε . The transfer function $f(r, z)$ involves the matrix element, which exponentially decreases with the distance between tip and sample.

From equation (1), we can deduce the expression of dI/dV spectrum:

$$\frac{dI}{dV}(r, \varepsilon) \equiv g(r, \varepsilon) = f(r, z)N(r, \varepsilon)e = \frac{eI_s(r, z, V_b)}{\int_0^{eV_b} N(r, \varepsilon) d\varepsilon} N(r, \varepsilon) \quad (2)$$

Setup condition can be normally determined by the bias voltage V_b and setpoint current I_s . With the same V_b , the larger I_s corresponds to a shorter distance between tip and sample. By fixing V_b and changing I_s , we found that the nematic-order signal (difference of dI/dV spectrum at Fe₁ and Fe₂ sites) is stronger when the tip is closer to the sample surface. When the tip is far away from the surface, the nematic-order signal is too small to be detected from the dI/dV spectrum. This phenomenon will help the understanding of following examples.

Although the tip is carefully prepared with the same procedure, the tip conductance may vary from tip to tip, which means that $f(r, z)$ is tip dependent. Occasionally, the variation of a special tip is relatively large. Then even with the same V_b and I_s , the tip-sample distance is different and

the nematic-order signal can show a different magnitude. We present such an example in Figs. S5p-t. Although the same setup condition $V_b = 100$ mV, $I_s = 100$ pA is applied for this tip, the magnitude of nematic order in Figs. S5s is four times smaller than that under general conditions, like in Fig. S5d.

In Figs. S5a-e, the tunneling condition is $V_b = 100$ mV, $I_s = 100$ pA. For comparison, a different tunneling condition, $V_b = 20$ mV, $I_s = 10$ pA, is used for data in Figs. S5f-j. From the nonlinear I - V relation (inset of Fig. S5f), we could infer that the tip-sample distance in Figs. S5f-j is shorter than that in Figs. S5a-e, although the ratio of V_b/I_s is even larger for Figs. S5f-j. Correspondingly, the magnitude of nematic order in Fig. S5i is almost two times larger than that in Fig. S5d.

For another comparison, a tunneling condition, $V_b = -20$ mV, $I_s = 20$ pA, is used for data in Figs. S5k-o. Here the tip-sample distance is comparable with that in Figs. S5f-j. In STM measurements, the dI/dV spectrum is normalized by the setup condition, keeping the integration of $N(r, \varepsilon)$ from 0 to eV_b the same at different positions. In Fig. S5i, the nematic order at -20 mV is already up. Then with $V_b = -20$ mV, the dI/dV spectra at Fe₁ and Fe₂ sites are relatively shifted by adjusting the tip-sample distance, to be compatible with the normalization requirement. Correspondingly, the nematic order in Fig. S5n is suppressed. However, the trend of the nematic order is still similar to that in Figs. S5d and S5i. Although we cannot define an absolute normalization condition in the dI/dV measurement, one reasonable approach is to set the bias voltage at which the nematic order is close to zero. At 77 K, the measurement with $V_b = 20$ mV, $I_s = 10$ pA is not very stable, and we have limited datasets with this junction. In comparison, $V_b = 100$ mV, $I_s = 100$ pA is a good choice of setup condition for measurement at 77 K.

With the same special tip, two datasets are taken with $V_b = 100$ mV, $I_s = 100$ pA (Figs. S5p-t), and $V_b = -100$ mV, $I_s = 100$ pA (Figs. S5u-y), respectively. This tip leads to a weak signal of nematic order at -100 meV (Fig. S5s). The setup condition of $V_b = -100$ mV, $I_s = 100$ pA roughly corresponds to a tunneling with $V_b = 100$ mV, $I_s = 50$ pA. The larger tip-sample distance mainly leads to the slightly suppressed nematic order in Fig. S5x compared with that in Fig. S5s. The weak nematicity at -100 mV makes the normalization a small effect.

In Fig. S8, we show two datasets taken in the same area but under different setup conditions at 4.5 K. The bias dependence of the nematicity can be reproduced under different bias voltages as well. The nematic order is around zero at both $V_b = 40$ mV and $V_b = 20$ mV. The trend of the nematic order is not affected by the setup conditions. With the principle discussed above, we prefer to choose a bias voltage within the gap. At 4.5 K, the measurement with $V_b = 20$ mV, $I_s = 20$ pA is stable, different from that at 77 K. The difference may originate from the variation of conductance of the FeSe sample at two temperatures.

* does analysis of maps taken with different density of points in the same area yield the same results on the [110] nematicity?

As discussed above about the pixel density, Figs. S5f, S5p and S5u are taken with 200×200 pixels for areas of 20 nm \times 20 nm. In Fig. S5a, the dataset is taken over a 12 nm \times 12 nm area with a larger pixel density of 160×160 . The trend of the [110] nematicity is robust against the density of pixels, which indicates that a pixel density of 10 pixels per 1 nm is enough for a reliable analysis. We also tested analysis of maps taken with different density of points in the same area, and obtain the similar trend of [110] nematicity as expected.

Reviewer #2 (Remarks to the Author):

The authors observed two different nematic orders in an Fe-based superconductor FeSe using scanning tunneling microscopy and spectroscopy at 77 K and 4.5 K. The anisotropic electronic states have been widely observed in the various unconventional superconductors. In general, the nematicities (B_{1g}) accompanied with collinear AFM spin order are developed along the [100] direction in Fe-based superconductors. Here, they observed the long-ranged B_{2g} nematicity along the [110] at 77 K. The coherence length of B_{2g} is smaller while the B_{1g} is enhanced at 4.5 K.

As the authors pointed out, the nematicity in unconventional superconductors has been intensively investigated with various approaches. Nonetheless, we haven't fully understood the origin of the nematicity and its relation with superconductivity. The coexistence of the B_{1g} nematicity and the collinear AFM spin order have been widely observed in the Fe-based superconductors. In the case of absence of spin ordering, the nematicity along different direction has been recently observed in the heavily doped Fe-base superconductors. Therefore, authors consider that the paramagnetic bulk FeSe is one of the candidates that has different nematicity from B_{1g} and provide a good platform to study the origin of the nematicity.

Compared with importance of their message, the data and analysis seems not sufficient to back up the strong claims made in their discussion. Unless the authors can appropriately rebut the comments below, my evaluation would be that the results, while interesting, are insufficient for publication in Nature Communications.

We thank reviewer#2 for the understanding of our purpose to search the [110] nematicity in bulk FeSe. We also thank reviewer#2 for recognizing the interest of our work. Below are our replies to the reviewer's concerns.

First, the authors visualize the B_{2g} nematicity along the [110] direction in the surface of FeSe sample using spatially resolved STS at 77 K (Fig. 2). It is different from the B_{1g} nematicity along the [100] direction that has been observed in this material. They used two different approaches to quantify the B_{2g} nematicity. One is acquiring the difference of the spectra of two different Fe sites and another comparing two FFT spectral intensities at the reciprocal lattice points corresponding to Se lattice. They seem to show strong anisotropic features along the [110] direction. In order to be more convincing to the reader, the authors should

1) Provide evidence that their tip is sufficiently isotropic.

In Supplementary note 2, we introduce the detailed method of tip optimization in our experiments. Tips are prepared by soft contact with an Au (111) surface. The sharp round tip is judged by the state-of-the-art topography of a single Au atom, as shown in Fig. S2a. The shape of the dumbbell defect is very sensitive to the tip apex. Any feature on the tip apex is conspicuous at the dumbbell sites in the topography. We can judge the tip condition by the topography of the dumbbell defect. To ensure an isotropic tip used for our experiment, we also take images including the dumbbell defects after finishing the grid spectroscopy.

Tips are prepared and judged by the same method at 4.5 K. At this temperature, the advantage is that there are twin boundaries, on two sides of which the [100] nematic order are 90° rotated (or

opposite) to each other. The detection of this 90° rotation proves that the [100] nematicity is the intrinsic property, instead of a phenomenon induced by an anisotropic tip. At 4.5 K, we also observe domains of local [110] nematic order with different signs, as shown in Fig. 4 and Figs. S6 and S7. Because the map in one image is taken by the same tip, the appearance of opposite nematic order on two sides of a domain wall rules out the possibility that the symmetry breaking is caused by the tip anisotropy [Science 344, 612 (2014)].

2) Show the acquired dI/dV spectra on Fe1, Fe2, Se atoms.

In Fig. S3, we show the averaged dI/dV spectra of Fe₁ and Fe₂ atoms, together with the averaged spectra of two groups of random Fe atoms. In Fig. S5, we show additional datasets. The [110] nematic order is reproducible at different setup conditions.

3) Present data acquired with $V_B = -100 \text{ mV} < 0$ and $I_{\text{set}} = 100 \text{ nA}$ or reasonable explanations to exclude the set-up effect. The larger variations in the negative bias seem to originate with the positive bias set-up.

As our reply to reviewer#1, we give a reasonable discussion and explanation about the setup effect in the measurement of nematic order.

4) In addition, I can see the impurities (weakly darker and brighter contrast) in Fig. 2a and the suppressed nematic order in Fig. 2b. even though they intentionally choose the field of view without defects to investigate intrinsic electronic properties. Do they have any correlation between impurity features in the STM images and local suppression in Fig. 2b? What is the origin of local suppression of B_{2g} nematicity?

The weak dark and bright spots in Fig. 2a may be due to the defects beneath the top FeSe layer, which has little effect on the analysis because the dI/dV spectra are mainly contributed by the top layer FeSe unit cell. Due to the small oscillation of Se atoms and the small scale we choose to show a clear lattice, these spots are amplified. The local suppressed order in Fig. 2d (not 2b) are not real physical features. They appear due to the mismatch between the grid pixels of local anisotropy map (1 pixel per unit cell) and the grid pixels of the frame.

Second, the authors quantify the B_{1g} nematicity as a function of bias using spatially resolved STS at 4.5 K (Fig. 3). The unidirectional feature close to the defect as an evidence for B_{1g} nematicity is consistent with previous STM studies on FeSe. STM images of two different domains and their anisotropic intensities of their FFT spectra also support the existence of the B_{1g} nematic order. They also did similar analysis for dI/dV maps as a function of bias. The higher bias show the stronger nematicity. In order to make their statements more concrete, the authors should

5) Compare the acquired dI/dV spectra with previous STS study of this material.

In the revised Fig. S1, we show the resistance curve and the dI/dV spectra acquired at 4.5 K. The superconducting critical temperature and the superconducting gap ($\Delta = 2.5 \text{ meV}$) are consistent with the published literatures about bulk FeSe [Nature Mater. 15, 159 (2016), Phys. Rev. X 5, 031022 (2015), Sci. Rep. 7, 44024 (2017)]. Because previous literatures are normally focused on the superconducting properties, a small voltage range is often chosen, e. g. within $\pm 20 \text{ mV}$. We are interested in the nematic order and choose a large voltage range in our measurement. The dI/dV

spectra in a large voltage range is consistent with a recent published work [Phys. Rev. Lett. 122, 077001 (2019)]. We tested different samples by different tips and feel confident about the quality of our samples.

6) Put the set-up conditions of dI/dV measurements and compare results acquired with two different polarities of set-up bias?

As in our previous discussion about the setup condition, we choose $V_b = 20$ mV, $I_s = 20$ pA for the measurement at 4.5 K. When considering the [100] order, the bias $V_b = 20$ mV is close to the Fermi level. The [100] order is weak at this bias voltage, and comparable with that at -20 mV (Fig. 3). We think that the bias polarity will not affect the main feature of this nematic order.

We also checked the [100] order measured with $V_b = 20$ mV at 77 K, which is negligibly small in our limited datasets. This is consistent with that the unidirectional stripes of dumbbell defects are clearly shown in Fig. 3a but missing in Fig. 1b, with both topographies taken under $V_b = 100$ mV, $I_s = 20$ pA. Because the [100] order can be detected at 77 K using other experimental techniques, e. g. the ARPES and NMR, the negligibly small [100] order at 77 K may be limited by the junction condition.

7) Explain why the higher bias voltages show the stronger nematicity and the nematicity is more clear at the negative bias.

A physical picture is that the nematic order is coupled in the dI/dV spectrum, from which we detect an energy-dependent *response* to the nematic order. Later in the reply to reviewer #3, we give a detailed discussion about the energy-dependent response. At the present stage, we focus on providing reliable and reproducible results, and cannot explain the quantitative dependence yet. Further advanced theorem should be developed to explain how the nematic order is coupled in the dI/dV spectrum, which will help to answer the energy-dependent response.

8) Put the length scale of the dashed lines in Fig. 3a and explain what they represent.

The unidirectional stripes straddling each dumbbell defect is a universal phenomenon in FeSe. As presented in a previous work [Phys. Rev. Lett. 109, 137004 (2012)], the length scale of the two parallel depressions under low bias voltage is approximately $16 a_{Fe-Fe} \sim 4.4$ nm. This is consistent with our results, as shown in Fig. R2a. Under a large bias voltage, the parallel depressions turn to be bright swallow tails, as shown in Fig. R2b as well as the published literature [Phys. Rev. X 5, 031022 (2015)]. The dashed lines are just for indicating the direction of the bright tail features, with the length scale roughly 8 nm long.

The unidirectional pattern under the large bias voltage is rotated by 90° with that under a lower bias voltage (Fig. R2, red dashed boxes). The unidirectional pattern is most likely originated from the quasiparticle scatterings between the d_{yz} orbitals of Fe atoms [Science 357, 75 (2017), Nat. Mater. 869 (2018)]. With d_{yz} orbitals composed along the k_x direction in the α band of the BZ center, the scattering pattern is along the a axis under the low bias voltage. With d_{yz} orbitals composed along the k_y direction in the ε band of the BZ corner, the scattering pattern is along the b axis under the large bias voltage. Here we use the same notations of axis with ref. [Science 357, 75 (2017). Nat. Mater. 869 (2018)] for the iron lattice.

In the revised manuscript, we added the explanation of this unidirectional stripe with the sentence ‘For the same dumbbell defect, the unidirectional pattern under the high bias voltage is rotated by

90° compared to that under the low bias voltage. This could be because that the unidirectional patterns are possibly originated from the scattering between the d_{yz} orbitals of Fe atoms [41]’.

Fig. R2 **a** A 70 nm × 70 nm STM image taken under $V_b = 20$ mV. **b** Roughly the same area with **a**, a 80 nm × 80 nm STM image taken under $V_b = 100$ mV. The red dashed boxes highlight the pattern of the same dumbbell defect under different bias voltages.

9) Did they normalize the FFT spectral intensities corresponding to B_1g nematicity by using the reference value (for example, FFT spectral intensities for Se lattice) to plot order parameter as a function of bias voltage?

We do not normalize the FFT spectral intensities, which are raw data extracted from dI/dV maps. Because both [100] and [110] nematicity exist, no such reference peaks are available for the normalization. The procedure of data collection also guarantees that we do not need such a normalization as how the nematic order is obtained in cuprate superconductors.

Third, the authors quantify the B_2g nematicity as a function of bias using spatially resolved STS at 4.5 K (Fig. 4). Contrast to results at 77 K, the negative and positive nematicity coexist and their strength is not zero at positive bias. The order parameter as a function of bias show the gap-like feature whose size is comparable to the gap of Neel spin fluctuation. In this section, the authors should

10) Show the acquired dI/dV spectra on Fe1, Fe2, Se atoms for positive, negative, zero order parameter region, respectively. Compare their spectra with ones acquired at 77 K.

In Fig. S8, we show the averaged spectra of Fe₁ and Fe₂ as well as the averaged spectra of two groups of random Fe atoms. The spectra at 4.5 K is overall consistent with that at 77 K. In Fig. S8f, we show a dataset with a larger energy range. The difference between the averaged spectra of Fe₁ and Fe₂ is visible to the naked eyes.

11) Check the possibility that the asymmetric feature in the order parameter spectra come from the set-up effect. If they have data acquired with opposite polarity set-up bias, compare the results with presented one in the manuscript.

As our reply to the reviewer#1, we give a reasonable discussion and explanation about the setup

effect in the measurement of nematic order. In Fig. S8, we show additional two datasets taken at the same area with different setup conditions at 4.5 K. The near [-50, 50] meV gap is robust under different bias voltages. At the present stage, we focus on providing reliable and reproducible results, and cannot explain the asymmetric feature yet. Instead, we emphasize the qualitative feature of a gap in the energy-dependent [110] nematic order.

12) Explain the origin of energy dependency of domain structure shown in Fig. S5?

In the revised manuscript, the previous Fig. S5 has been changed to Fig. S6. In Fig. 4, the collective order is opposite with different signs of domains. With a small value of $|E|$ ($|E| < 50$ meV), the local anisotropy map is rambling with disorganized domains. The value of the local anisotropy with small $|E|$ is one magnitude smaller than that at high $|E|$ (Figs. S6e-h). Eventually, the collective order at low $|E|$ is nearly zero. With the increase of the $|E|$, two domains with opposite signs of nematicity are developed. The pattern is almost not changed with further increase of the energy, similar to the evolution of local anisotropy pattern in cuprate superconductor [Sci. Rep. 7, 8059 (2017)]. When the order is well developed, the local anisotropy is relatively homogeneous within each domain.

13) Explain why the negative order parameter seems to have larger and clear gap than the positive one in Fig. 4b?

We are sorry for this confusion. Please note that the order parameter is an averaged result in the domain area. The ‘larger’ and clearer gap for the negative nematic order is related with the stronger signal of the collective nematic order (blue area). In Fig. S7, the gaps for the [110] nematic order are consistent with that in the main text. In Fig. S8, the [110] order without domains is more robust, the strong signal leading to the clean gap feature in Figs. S8d and S8i. In the revised manuscript, we added ‘The clearer gap in Fig. 4d is due to the averaging of stronger signals in this domain’ to avoid this confusion.

Reviewer #3 (Remarks to the Author):

The others report an STM study of clean FeSe single crystals at liquid nitrogen (77K) and liquid He (4.5 K) temperatures. Their main novel finding is a difference in the differential conductance between “Fe1” and “Fe2” sites, where the Fe1 and Fe2 sites are arranged in an alternating fashion. This difference has been dubbed “real space B2g nematic order” and its value is strongly energy dependent and also temperature dependent.

The data appear to be of high quality and the manuscript is well organized and – for the most part – clearly written. However, many questions remain open.

The so-called Fe1 and Fe2 sites are distributed in a checkerboard like manner. In consequence, a difference between the averaged differential conductance at sites of Fe1 and Fe2 atoms does, as far as I can see, not imply a *nematic* order, which is defined as a rotational symmetry breaking. The authors write “Here we note that each type of Fe atoms are rotated by 45° with the [100] direction of the Fe lattice (Fig. 1a). The difference between the differential-conductance at Fe1 and Fe2 sites thus represents an electronic anisotropy along the [110] direction, which has a form of B2g

symmetry.” This reasoning is unclear to me. Indeed, the authors discuss how this exotic order might couple to Neel-type magnetic fluctuations (which also are not nematic). What makes the authors choose the term “B_{2g} nematic” for this real-space order?

We agree with the reviewer that Fe₁ and Fe₂ form a checkerboard-like pattern if we only consider the Fe plane. In our paper, the rotational symmetry breaking along [110] direction is defined including additional Se atoms. This definition is enlightened by the similarities between the crystal structure of iron-based superconductors and that of cuprate superconductors, which has been mentioned previously in ref. [Nat. Commun. 5, 5761 (2014)]. As shown in Fig. R3, the top-layer Se atoms and Fe atoms can be mapped to the CuO plane of cuprate superconductors. In a cuprate superconductor, the inequivalent differential conductance at O_x and O_y atoms in CuO unit cell leads to a [100] nematic order with respect to the Cu-Cu lattice direction [Nature 466, 347 (2010)]. In our experiment, the inequivalent differential conductance at Fe₁ and Fe₂ atoms are determined, which is similar to that in the cuprate superconductor. The inequivalent direction turns to be [110] direction with respect to the Fe-Fe lattice. Correspondingly, the conductance map must be centered at a top Se atom before the Fourier transform, similar to the nematic order in cuprate superconductor which depends on Fourier transform of a conductance map centered at a Cu atom [Nature 466, 347 (2010)]. Both the analysis method and the results are similar to those in the STM study of intra-unit-cell nematic order in the cuprate superconductor [Nature 466, 347 (2010)]. Then we inherit the definition and also call this rotational-symmetry-breaking a *nematic order*. The Neel-type magnetic order is often not related with a nematic order, because only the Fe plane is considered. The checkerboard-like pattern of spin is however similar to the inequivalent Fe₁ and Fe₂ atoms we discussed.

We note that nematicity defined in some experiments is not related with a rotation axis, and the definition of nematicity can be sometimes puzzling. To avoid the possible confusion by different terms, in the revised manuscript we define the inequivalent Fe₁ and Fe₂ atoms along the [110] direction as [110] nematicity, without referring to the B_{2g} nematicity. For the sake of clarity, we also define the inequivalent Q_a and Q_b along the [100] direction as [100] nematicity, without referring to the B_{1g} nematicity. We replace the corresponding sentence with ‘The top layer Se and Fe atoms can be mapped into the CuO plane of a cuprate superconductor. The sites of Se (Fe) atoms can be mapped one by one into the sites of Cu (O) atoms (Fig. 1a). In the cuprate superconductor, an inequivalent differential conductance between two types of oxygen atoms leads to a nematic order along the Cu-Cu lattice direction. The difference between the differential conductance at Fe₁ and Fe₂ sites thus represents a symmetry breaking along the [110] direction of Fe-Fe lattice’.

Fig. R3 Schematic top view of the CuO plane of a cuprate superconductor and FeSe plane of FeSe

superconductor. The black dots represent the axis of rotation perpendicular to the plane of paper.

In the bulk crystals, “Fe1” and “Fe2” sites are symmetry equivalent. What is the role of the surface in breaking the symmetry between the “top Se” and “Inner Se” atoms? May this have an effect on the observed unusual symmetry breaking?

The chemical environment for two layers of Se atoms in a FeSe layer is equivalent for the FeSe layer in the bulk. For the top FeSe layer which is exposed for STM experiment, the chemical environment may be different for the top and inner layer of Se plane. Because adjacent two layers of Se planes are connected by van der Waals’ interaction, the effect of two inequivalent Se layers on Fe lattice is considerably small. Despite the small difference between the averaged dI/dV spectra of Fe₁ and Fe₂, the order parameters for [110] and [100] nematic orders extracted from the same dataset are at the same level (Figs. 3 and 4). The latter is a well-known intrinsic nematic order of the bulk FeSe. The trade-off evolution of the two nematic orders with temperatures also indicates that the two have comparable magnitudes. Besides, we observe domains of local anisotropy of [110] order at low temperature. The appearance of different signs of domains excludes the possibility of a surface effect, because the top layer Se atoms are exposed equivalently to the vacuum.

What are the implications of an energy-dependent order parameter of a static order? Do the authors have a traditional Landau-order-parameter framework in mind, as is often the case in the field?

We do agree that the nematic order itself does not evolve with the energy, as a traditional order described by the Landau-order-parameter. Here the energy dependent behavior is a special phenomenon we discuss in the field of STM. The energy-dependence means the bias-voltage-dependence in the dI/dV spectrum. No matter how an order is coupled in the dI/dV spectrum, the dI/dV spectrum may have an energy-dependent *response* to the static order. Sometimes the coupling is energy dependent. For example, the charge order has been detected in dI/dV maps of cuprate superconductor Bi₂Sr₂CuO_{6+x}. In dI/dV maps at different energies, the Q vector of charge order is non-dispersive with energy, while the intensity of the charge order is energy dependent, showing a peak around the pseudogap energy [SCPMA 63, 227411 (2020)]. A similar energy dependent trend has also been observed for the nematic order in the STM study of Bi₂Sr₂CaCu₂O_{8+x} [Nature 466, 347 (2010)]. On the other hand, because the dI/dV spectrum in STM is averaged for k -points with different directions but the same energy in the k -space electronic bands, the dI/dV spectrum can have an energy-dependent sensitivity to different parts of the k -space bands. Then a momentum-dependent order can result in an energy-dependence in the dI/dV spectrum.

In our experiment, the [-50, 50] meV gap feature of the [110] nematic order is specially found at low temperature. Considering a similar gap in the Neel spin fluctuations at low temperature, we presume that the nematic order in the dI/dV spectrum is related with an energy-dependent coupling of the Neel spin order. Further advanced theorem should be developed to explain the complex coupling, and how the fluctuations is partially pinned down as a static order. It is also a general question to explore how the static Neel magnetic order is coupled in dI/dV spectrum, like the origin of nematic order in Bi₂Sr₂CaCu₂O_{8+x} [Nature 466, 347 (2010)].

The dI/dV spectra are of primary importance for the energy dependence of the effect. The authors should show a representative selection of them, at least in the supplement.

In the revised supplementary material, we show dI/dV spectra in Figs. S1, S3, S5 and S8, for measurements both at 77 K and 4.5 K. The spectra are consistent with the published literatures about bulk FeSe. In the revised manuscript, we also added more datasets and the corresponding dI/dV spectra. The difference between averaged spectra at Fe₁ and Fe₂ can be reproduced both at 77 K and 4.5 K.

In summary, the data seem very nice and potentially also very interesting. However, these open questions make it difficult for me to recommend publication.

REVIEWER COMMENTS

Reviewer #1 (Remarks to the Author):

The authors have taken on board many of the comments by the reviewers and provided additional data to support their claim. While the manuscript is strengthened, there remain a few points where the authors need to provide additional clarification before the manuscript can be accepted for publication:

(1) I am not convinced by their response with regards to the choice of the filter radius which is applied. Given that the local anisotropy looks completely different for different filter radii (compare supplementary figure 4), there needs to be some independent criterion for choosing which filter radius leads to a correct representation of the electronic states in the Fe1 and Fe2 positions. How do we know that the local anisotropy is not as shown in supplementary fig. 4a, b? The authors only refer to previous work in this regard where they chose a similar filter radius in the cuprates, but a key difference is that in the cuprates one can see the typical length scale of the nematic patterns in real space and therefore estimate what filter radius to choose from the real space patterns. On what basis is it chosen here? I do not think that using the one used to analyse nematicity in cuprates is appropriate, as it is not clear that the phenomena are related. Also, why does the local anisotropy look different at positive and negative bias in fig. S4c, d? For a static local anisotropy, one would have expected it to be the same at all energy scales (as appears to be broadly the case in fig. S6).

(2) Some of the topographies shown in the main manuscript as well as in the supplementary show dark lines running through the image (e.g. in figs. 2a, 3b,c, particularly noticeable in the new supplementary fig. S5 panels f, k, p, u) which are probably due to noise. Given the tiny differences in the signal which the authors discuss, can they exclude that such noise leads to artefacts in their analysis?

Reviewer #2 (Remarks to the Author):

In their Reply the authors provide plausible arguments and additional data to clarify the concerns of three referees. Especially, their supplementary materials are helpful to rule out some artifacts that probably attributed to the set-up and the anisotropic shape of the STM tip. However, their reply to comments about gap size of order-parameter spectra for the nematicity along [110] direction is still unclear to me. The lower signal (red curve) in Fig. 4b of the main text shows smaller (less clear) gap than the larger one (blue). As shown in their supplementary Fig 8, the larger signal spectrum has the smaller gap. This means that their dI/dV signal is not large enough to determine the gap feature. To clarify the gap and quantify its size, I would recommend to plot them with a logarithmic scale. The gap and its size will be informative to find a correlation between the nematicity and the Neel spin fluctuations as author mentioned in the Discussion section.

Until the concern raised above are addressed, I would support publication in Nature Communications.

Reviewer #3 (Remarks to the Author):

I appreciate the diligence with which the authors have responded to many of the criticisms. My question concerning the apparent energy dependence of the order parameter is resolved, thank you. I believe that the data and analysis are sound. However, I strongly object to naming the observed novel order “[110] nematic” or nematic in any way.

To explain: I appreciate Figure R3, which helps a lot to make my concern clear. I think the essential point here is that only “The top layer Se and Fe atoms can be mapped into the CuO plane of a cuprate superconductor”, but the inner Se layer is neglected. This can only be justified if – by some mechanism – the inversion symmetry that transforms the top Se layer into the bottom Se layer is broken. However, in their reply, the authors affirm that they do not think such a symmetry breaking (e.g., by the presence of a surface) is relevant “the effect of two inequivalent Se layers on Fe lattice is considerably small” and that “The appearance of different signs of domains excludes the possibility of a surface effect”. So, we cannot neglect the bottom Se layer. I now took the liberty to slightly edit Fig. R3 (please see attached figure).

I agree that the symmetry at the Se sites is *locally* broken. However, a 4-bar symmetry of 90deg rotation and inversion at each Fe site remains. There is a tetragonal unit cell (indicated in black), which is twice as large as it would be without the observed novel order, but importantly the system remains tetragonal. The connotation of the name “nematic” in the iron-based systems, such as FeSe, is breaking globally the tetragonal symmetry. Given this connotation, the authors’ naming of the observed novel order is highly misleading!

I do not call into question the data and the analysis itself. If the manuscript were rewritten so that the novel order is no longer called “nematic” (which obviously means substantial rewriting of some sections) I would support publication.

As a note, the observed novel order has the same symmetry as the charge order predicted to be the vestigial order to the so-called charge-spin density wave magnetic order in some iron-based systems, see Phys. Rev. B, 100, 014512.

Reply to reviewers:

Our replies are in blue color and the reviewers' comments are in black color.

Reviewer #1 (Remarks to the Author):

The authors have taken on board many of the comments by the reviewers and provided additional data to support their claim. While the manuscript is strengthened, there remain a few points where the authors need to provide additional clarification before the manuscript can be accepted for publication:

(1) I am not convinced by their response with regards to the choice of the filter radius which is applied. Given that the local anisotropy looks completely different for different filter radii (compare supplementary figure 4), there needs to be some independent criterion for choosing which filter radius leads to a correct representation of the electronic states in the Fe1 and Fe2 positions. How do we know that the local anisotropy is not as shown in supplementary fig. 4a, b? The authors only refer to previous work in this regard where they chose a similar filter radius in the cuprates, but a key difference is that in the cuprates one can see the typical length scale of the nematic patterns in real space and therefore estimate what filter radius to choose from the real space patterns. On what basis is it chosen here? I do not think that using the one used to analyze nematicity in cuprates is appropriate, as it is not clear that the phenomena are related. Also, why does the local anisotropy look different at positive and negative bias in fig. S4c, d? For a static local anisotropy, one would have expected it to be the same at all energy scales (as appears to be broadly the case in fig. S6).

Here we explain with more details to clarify the questions raised by the 1st reviewer. The goal of our experiment is to detect a weak signal, while the tiny difference between the conductance of each pair of Fe₁ and Fe₂ atoms in a single unit-cell is buried in unavoidable noise or signal fluctuations. In Fig. 2c, the collective order is obtained by subtracting the average dI/dV spectra of Fe₂ atoms from that of the Fe₁ atoms. Each of the average spectrum is averaged for more than two thousand times, so that the effect of signal fluctuations can be averaged out in the average spectrum. When we extract the local order, the order value in a single unit-cell is greatly affected by signal fluctuations, as shown in the local order maps [Supplementary Figs. 4a and 4b as examples]. We cannot obtain useful information from the noisy local maps to judge any local order physics.

After a careful comparison between q space and r space collective orders, we realize that we can choose a Gaussian filter to remove the effect of signal fluctuations. The standard procedure is to obtain the Fourier-transform of an r space map, use a Gaussian filter around related Q vectors, make a reversed Fourier-transform map of the q space filtered map, then calculate the final local order map. Figure R1 shows the local order maps and local order distributions obtained with different filter radius. Local order values in each map are consistent with a Gaussian distribution, the center of which is almost the same as the averaged collective order [Figs. R1d, R1f and R1h]. With the filtration, signal fluctuations of the local order pattern can be largely suppressed, which is also reflected in the substantial decrease of the distribution width [Figs. R1d, R1f and R1h]. The filtered local order map can be understood as that each single point value is averaged over an area of the filter size. The criterion for a critical filter size is that the obtained local order patterns would not change with further increase of the filter radius [also see Supplementary material in Sci. Rep. 7, 8059 (2017)]. In Fig. R1, once the filter radius exceeds $\Lambda^{-1} = 1$ nm, both the local pattern and the width of the Gaussian distribution are nearly unchanged. Thus, a filter radius of 1 nm is a proper

choice to get a clean local order pattern. Although similar to the value used for the nematic order in cuprates, the filter radius is independently determined for FeSe with the above criterion. In the local order map with well-developed order [Supplementary Fig. 4c], the map shows strong order signals (with red color) across the whole area. In the local map whose collective order is around zero, the map shows weak order signals with clear domains (red and blue) and domain walls [Supplementary Fig. 4d]. Only in the filtered local map, signal fluctuations are locally suppressed, then clear domain and domain walls appear. In this filtered local order map, domains larger than 1 nm will not be affected by the filtration while features smaller than 1 nm is physically insignificant (like a tiny island smaller than 1 nm). We explained the meaning of the filter size in more detail in the revised Supplementary Note 4.

We note that the local order pattern without the Gaussian filtering in the cuprate experiment is also irregular like that in FeSe [see Supplementary material in Sci. Rep. 7, 8059 (2017)]. The clear pattern can be observed only with the signal fluctuations removed [Nature 466, 347(2010), Science 344, 612 (2014), Sci. Rep. 7, 8059 (2017)]. Some other electronic orders such as the smectic order (sometimes called a charge order) in cuprate superconductors is visible to the naked eyes [Fig. 4E in PNAS 111 (30), E3026 (2014)]. Although the much stronger smectic order has relationship with the nematic order, they are two different orders with different Q vectors.

Fig. R1 Local order maps and order value distributions with different filter radius. Scale bars are 2 nm. The filter size is shown by a black circle in each filtered local order map.

The collective [110] electronic order (to avoid the possible misleading by the terms ‘[110] nematic order’, we substitute the ‘[110] nematic order’ with the ‘[110] electronic order’ as suggested by reviewer #3) at 77 K shows a trend in Fig. R2a. The collective order at -100 meV is a finite value, as indicated by the green arrow in Fig. R2a. Correspondingly, the local order map is robust at this energy, shown with a relatively large magnitude across the whole map [Fig. R2b]. With the increase of energy, the collective order decreases continuously. The value of the collective order already approaches zero at 100 meV, as indicated by the black arrow in Fig. R2a. Consequently, the local order map at 100 meV shows a pattern of small domains with different signs [Fig. R2c], the average value of which is close to zero and consistent with the collective order. At the same time, the value of the local order reduces to an order of magnitude smaller than that at the negative energy [see the scale bar in Fig. R2c].

The collective charge order at 4.5 K shows a different trend from that at 77 K [Fig. R2d]. The

field of view of Fig. R2e can be roughly divided into two domains with different signs. The collective orders for domains with different signs are finite values at high $|E|$, as indicated by the purple and light blue arrows in Fig. R2d. Correspondingly, the local orders show static (or well-developed) patterns at these high absolute energies [Figs. R2e and R2g]. Although some pudding-like blue areas appear in the upper left domain in Fig. R2g, they have little effect on the overall collective order for the upper left domain. The collective order at low $|E|$ is almost zero, as indicated by the yellow arrows in Fig. R2d. Consequently, the pattern at low $|E|$ shows a pattern of domains with different signs [Fig. R2f], the average value of which is close to zero and consistent with the collective order. At the same time, the value of the local order reduces to an order of magnitude smaller than that at high $|E|$ [see scale bar in Fig. R2f].

Altogether, the evolution of the energy dependent local order maps of FeSe is consistent with the trend of the collective order. The local order grows with the increase (or decrease) of the energy. When the collective order is large enough, the local order pattern pins down and doesn't evolve with further increase (or decrease) of the energy, indicating a well-developed order parameter above this energy.

In our previous reply to reviewer #3, we have explained that the energy dependent behavior of the electronic order is a special phenomenon we discuss in the field of STM. A static order itself does not evolve with the energy, as a traditional order described by the Landau-order-parameter. The energy-dependence in STM means the bias-voltage-dependence in the dI/dV spectrum. No matter how an order is coupled in the dI/dV spectrum, the dI/dV spectrum may have an energy-dependent *response* to the static order. Some examples and possible mechanisms are discussed in the previous rebuttal letter.

Fig. R2 **a** Collective order at 77 K reproduced from Fig. 2c. **b-c** The maps of local order patterns reproduced from Supplementary Figs. 4c and 4d. **d** The collective order at 4.5 K reproduced from Fig. 4b. **e-g** The maps of local order patterns reproduced from Supplementary Figure 6. The black dashed lines in **c**, **e**, **f** and **g** highlight the boundaries between two domains with different signs.

(2) Some of the topographies shown in the main manuscript as well as in the supplementary show dark lines running through the image (e.g. in figs. 2a, 3b,c, particularly noticeable in the new

supplementary fig. S5 panels f, k, p, u) which are probably due to noise. Given the tiny differences in the signal which the authors discuss, can they exclude that such noise leads to artefacts in their analysis?

The periodic dark stripes in the topographies are indeed due to the unavoidable noise. However, this doesn't mean a terrible noise environment for our experiments. In fact, the overall noise background for the experiments is within an amplitude of 10 fm (*Z*-spectroscopy). It is the small amplitude of the atomic oscillation (see the color bars of the topographies) of FeSe that makes this small noise observable. In our experience, the atomic resolved topography can only be achieved under a good condition of the instrument with rather low noise level, especially at 77 K.

We use Lawler-Fujita algorithm to precisely locate the positions of each atom. The thermal drift can be removed after the drift correction. The Bragg peaks shrink into one pixel after this process, while the peaks of the noise fluctuations do not collapse into one pixel [Nature Materials 11, 585 (2012)]. As shown in Fig. R3, the noise peaks show blurred patterns after the drift correction. With the separation of *Q* vectors, the calculated precise location of atoms would not be affected by these alien peaks, similar to that in Ref. [Nature Materials 11, 585 (2012)].

After precisely locating each atom site, we extract dI/dV spectra of Fe₁ and Fe₂ atoms, respectively. The average spectrum for Fe₁ and Fe₂ atoms is thus respectively calculated. Due to the different phase of the noise on each raw spectrum, the noise fluctuations can be averaged out after averaging for enough times. The average spectra for Fe₁ and Fe₂ atoms are with a reasonable signal to noise ratio. The calculated [110] electronic order is reliable. In fact, we have clarified that the [110] electronic order deduced from *q* space is quantitatively consistent with that deduced from *r* space [Fig. 2 in the main text]. The inequivalent intensity of *Q_x* and *Q_y* (with Fourier transform centered at a site of Se atom) represents the inequivalent conductance of Fe₁ and Fe₂ atoms. The wave vectors of *Q_x* and *Q_y* are not related with the long-range noise fluctuations. As shown in Figure R3, there are no any overlaps between the dashed circles (*Q_x* and *Q_y*) and solid ellipses (long range noise peaks).

We feel confident about the conclusions we made regardless of the noise. Firstly, the datasets were taken at different times with different noise levels, and the results are robust against the variables such as time and noise fluctuations. Secondly, the main idea of the Lawler-Fujita algorithm is to aggregate the small physical signal by averaging the spectra for many times. The effect of the noise for a collective order is thus greatly reduced. Thirdly, the [110] electronic order comes from the inequivalent conductance of Fe₁ and Fe₂ atoms, which has the same wave vector with the lattice constant of Se atoms. The long-range noise in the topographies have wave vectors for several nanometers (black solid ellipses in Fig. R3), which can be removed by the Gaussian filter applied at *Q_x* and *Q_y* in the analysis of local order patterns.

We added a statement about the effect of the noise pattern in Supplementary Note 5:

Weak noise patterns are visible in Supplementary Figures 5f, 5k, 5p and 5u, due to the small oscillation of the Se atoms. However, these long-range stripes (with wavelength of several nanometers) have little effect on the obtained [110] order, because there are no overlaps of wave vectors between noise peaks and Bragg peaks (Supplementary Note 4).

Fig. R3 Topographies reproduced from Supplementary Figures 5f, 5k, 5p and 5u, and their corresponding Fourier transform maps after the drift-correction. The blue and red dashed circles indicate the Bragg peaks (Q_x and Q_y) of the Se lattice. The black solid ellipses indicate the long-range noise peaks.

Reviewer #2 (Remarks to the Author):

In their Reply the authors provide plausible arguments and additional data to clarify the concerns of three referees. Especially, their supplementary materials are helpful to rule out some artifacts that probably attributed to the set-up and the anisotropic shape of the STM tip. However, their reply to comments about gap size of order-parameter spectra for the nematicity along [110] direction is still unclear to me. The lower signal (red curve) in Fig. 4b of the main text shows smaller (less clear) gap than the larger one (blue). As shown in their supplementary Fig 8, the larger signal spectrum has the smaller gap. This means that their dI/dV signal is not large enough to determine the gap feature. To clarify the gap and quantify its size, I would recommend to plot them with a logarithmic scale. The gap and its size will be informative to find a correlation between the nematicity and the Neel spin fluctuations as author mentioned in the Discussion section.

Until the concern raised above are addressed, I would support publication in Nature Communications.

We thank reviewer #2 for the suggestion. The gap size roughly determined from Fig. 4b is not very precise. The gap of the red curve seems to be smaller than that of the blue curve in Fig 4b. This is due to the weaker signal of the red curve as well as the smaller y-axis range applied to the red curve. In the case of Supplementary Figure 8i, the signal to noise ratio is larger, and the non-zero order becomes more prominent than that in Fig. 4b. As a result, the gap seems to be smaller than that of the blue curve in Fig. 4b. In Fig. R4a, we plot the curves in Fig. 4b and Supplementary Figure 8i together. We also plot the differential curves of the order parameters, as shown in Fig. R4b. The differential curves with gapped order parameters should be zero. Based on this criterion, we judge that the gap size of the [110] electronic order at 4.5 K is around [-40, 40] meV.

We also plot all the energy dependent collective order measured at 4.5 K together, as shown in Fig. R4c. To have a better visualization, we intentionally invert the value for those negative order parameters, so that all the values are positive. We enlarge the gap area in Fig. R4c and plot the order

parameters with a logarithmic scale, as shown in Fig. R4d. Due to numerical errors, the gapped order is not absolute zero. The near-zero order under a logarithmic plot would have rather large fluctuations, while a finite order under a logarithmic plot would have small fluctuations. We can also judge the gap size based on the strength of the fluctuations of the curves in Fig. R4d. As indicated by the shaded blue area, the gap for the curves is roughly consistent with the knee points of the logarithmic curves. The gap size determined for the [110] electronic order in our experiment is comparable to that of the Neel spin fluctuations in INS experiment at 4 K, which has a gap size of roughly 35 meV [Nat. Commun. 7, 12182 (2016)]. We note that the gap size determined in this experiment and INS experiment is not a precise value due to limited resolutions in both experiments. The comparable gap size between different experiments suggests a likely correlation between this [110] electronic order and Neel spin fluctuations.

In the revised manuscript, we modified the estimated gap size to be around [-40, 40] meV.

Fig. R4 **a** Order parameters reproduced from Fig. 4b and Supplementary Figure 8i. **b** The differential curves of **a**. The shaded blue area indicates the gap range. **c** Order parameters in the manuscript at 4.5 K are plotted together. The values of the negative order parameters are intentionally inverted for a better visualization. **d** Logarithmic plot of the order parameters of the red box in **c**. The shaded blue area indicates the same gap size as that in Figs. R4a and R4b.

Reviewer #3 (Remarks to the Author):

I appreciate the diligence with which the authors have responded to many of the criticisms. My question concerning the apparent energy dependence of the order parameter is resolved, thank you. I believe that the data and analysis are sound. However, I strongly object to naming the observed novel order “[110] nematic” or nematic in any way.

To explain: I appreciate Figure R3, which helps a lot to make my concern clear. I think the essential point here is that only “The top layer Se and Fe atoms can be mapped into the CuO plane of a cuprate superconductor”, but the inner Se layer is neglected. This can only be justified if – by some mechanism – the inversion symmetry that transforms the top Se layer into the bottom Se layer is broken. However, in their reply, the authors affirm that they do not think such a symmetry breaking (e.g., by the presence of a surface) is relevant “the effect of two inequivalent Se layers on Fe lattice is considerably small” and that “The appearance of different signs of domains excludes the possibility of a surface effect”. So, we cannot neglect the bottom Se layer. I now took the liberty to slightly edit Fig. R3 (please see attached figure).`

I agree that the symmetry at the Se sites is *locally* broken. However, a 4-bar symmetry of 90deg rotation and inversion at each Fe site remains. There is a tetragonal unit cell (indicated in black), which is twice as large as it would be without the observed novel order, but importantly the system remains tetragonal. The connotation of the name “nematic” in the iron-based systems, such as FeSe, is breaking globally the tetragonal symmetry. Given this connotation, the authors’ naming of the observed novel order is highly misleading!

I do not call into question the data and the analysis itself. If the manuscript were rewritten so that the novel order is no longer called “nematic” (which obviously means substantial rewriting of some sections) I would support publication.

As a note, the observed novel order has the same symmetry as the charge order predicted to be the vestigial order to the so-called charge-spin density wave magnetic order in some iron-based systems, see Phys. Rev. B, 100, 014512.

We agree with the reviewer that the tetragonal unit cell with twice the iron lattice still preserves C_4 symmetry, although the symmetry of the intra unit cell for a single iron lattice is locally broken. Especially, we are convinced by the definition of the term ‘nematic’ in the reviewer’s comments:

The connotation of the name “nematic” in the iron-based systems, such as FeSe, is breaking globally the tetragonal symmetry.

To be consistent with the knowledge of nematicity and to avoid the possible confusion, we rename the observed novel order as a [110] electronic order (checkerboard charge order). This change also

does not affect the main experimental results and the discussion.

Correspondingly we have revised our manuscript. The main changes are abstracted as follows:

While most superconductors with [100] nematic order are accompanied with an antiferromagnetic phase transition, bulk FeSe doesn't form long range magnetic order at low temperatures. Instead, various spin fluctuations have been found in bulk FeSe. Spin fluctuations have been proposed as the origin of ordered states such as the nematic order. In the section of introduction, we revised our motivation for searching exotic electronic orders in FeSe besides the traditional [100] nematic order.

Since we do not call the [110] electronic order as a nematic order, we deleted the phase diagram (Supplementary Figure 9) which was concluded combined with our experiment and the former published [110] nematicity work. This will not affect the main content of the paper.

REVIEWER COMMENTS

Reviewer #1 (Remarks to the Author):

I appreciate the effort made by the authors to provide a thorough reply to my queries about the validity of the determination of the asymmetry between the Fe1 and Fe2 sites. Having read the reply by the authors, I remain sceptical of the validity of the analysis. This is mainly because the difference of the signal between Fe1 and Fe2 site is tiny (of order 10^{-3} , I assume in relative units) and becomes only detectable after Fourier filtering of the data and processing by a Lawler-Fujita algorithm. The argument by the authors, that, according to their fig. R1, they can chose a filter radius of 1nm^{-1} because for larger filter radius the patterns do not change anymore is weakened by the fact that the image does not show any variation already at 1nm^{-1} any more (i.e. it is completely red).

What would convince me of the analysis?

(1) demonstration that the same algorithm applied to data from a material which does not have a similar electronic order does not exhibit any evidence of a difference between the two sites corresponding to the Fe1 and Fe2 sites here, independent of filter radius. (for example ZrSiS?)

(2) demonstration that the same algorithm applied to simulated data (including drift, noise, applying the Fujita-Lawler algorithm, etc) would show no difference between what corresponds to Fe1 and Fe2 sites.

At the very least, before the manuscript is considered for publication, the authors should include the unprocessed real space differential conductance maps in the supplementary material, together with the conductance maps after processing.

Reviewer #2 (Remarks to the Author):

I think the authors have addressed very well of my concerns about the size of the energy gap. They showed data acquired on several different areas with logarithmic scales, which convincingly establish the energy gap of ~ 80 meV. However, a correlation between the [110] electronic order (determined by STM) and Neel spin fluctuations (determined by INS) is not trivial. An explanations of how to rule out that two independent phenomena exhibit similar energy scale features in their own spectra is still missing. Once the authors clearly address this issue, I would support publication in Nature Communications.

Reviewer #3 (Remarks to the Author):

The authors have amended the manuscript in response to my last comment in a satisfactory way. As far as I can judge, the manuscript is acceptable in Nature Communications.

Small note: Instead of "Phenomenally" (p.11), maybe the authors would like to write "Phenomenologically"?

Reply to reviewers:

Our replies are in blue, and the reviewers' comments are in black.

Reviewer #1 (Remarks to the Author):

I appreciate the effort made by the authors to provide a thorough reply to my queries about the validity of the determination of the asymmetry between the Fe1 and Fe2 sites. Having read the reply by the authors, I remain sceptical of the validity of the analysis. This is mainly because the difference of the signal between Fe1 and Fe2 site is tiny (of order 10^{-3} , I assume in relative units) and becomes only detectable after Fourier filtering of the data and processing by a Lawler-Fujita algorithm. The argument by the authors, that, according to their fig. R1, they can chose a filter radius of 1nm^{-1} because for larger filter radius the patterns do not change anymore is weakened by the fact that the image does not show any variation already at 1nm^{-1} any more (i.e. it is completely red).

We understand the reviewer's concern about the validity of the analysis. To confirm this weak 'hidden' electronic order, we carefully prepared the tip and excluded the effect of tip asymmetry. As shown in the manuscript and the previous reply letter, we have repeatedly measured and analyzed the order. Although the order magnitude is small, we can directly obtain the order versus energy trend without the filtering process. The collective order remains no change before and after the filtering process. The measured results at 77 K and 4.5 K are reproducible, and we could give a reasonable logic between the two results. We also reproduced and discussed the order measured by different tips, different samples, and under different tunneling conditions. We are then confident that the systematic results could not be from a faulty analysis.

It's reasonable to have a weak 'hidden' electronic order because the signal may reflect a magnetic order which is weakly coupled in the electronic channel. Lawler-Fujita algorithm with the Fourier filtering is not a new method, which has been proved useful in previous different applications. Here the algorithm is mainly used to detect the weak signal by averaging multiple measurements to increase the signal-to-noise ratio (a standard approach in weak signal measurement).

Here we also give a further explanation about the filter radius of 1 nm in previous Fig. R1. In our analysis, we put the same size of filter to all of the measured conductance maps. As shown in the new Fig. R1, we put the local order maps and the local order distributions of -100 meV and 0 meV together. The chosen radius is not a rigorously quantified result. From the filtered pattern, a filter radius of 0.5 nm has been a good enough value. We chose a more conservative value of 1 nm because the Gaussian distribution of filtered data is relatively stabilized for a larger radius.

What would convince me of the analysis?

- (1) demonstration that the same algorithm applied to data from a material which does not have a similar electronic order does not exhibit any evidence of a difference between the two sites corresponding to the Fe1 and Fe2 sites here, independent of filter radius. (for example ZrSiS?)

We understand the reviewer's consideration. Unfortunately, we don't have an appropriate dataset of ZrSiS for this analysis. Although we worked with ZrSiS sometime ago, our work's primary purpose was to study the defect-induced quasiparticle interference and we focused on the data around single defects instead of the clean-area data.

As our former reply to the reviewers and the experiment setup described in the supplementary material, to detect similar electronic order in a material is far from an easy job. A sharp isotropic tip

is needed, and the tip isotropy should be checked. Reproducible data should be confirmed for different tips and samples. The energy scale of measurement cannot be too small or too large. The logic connection among datasets under different conditions should be carefully considered. It requires much effort to prove whether another material exhibits a similar order, which is beyond the scope of this manuscript.

Fig. R1 **a-h** Reproduced from Fig. R1 in the last rebuttal letter. Local [110] electronic order and their distributions at -100 meV versus different filter sizes. **i-p** Local [110] electronic order and their distributions at 0 meV versus different filter sizes.

- (2) demonstration that the same algorithm applied to simulated data (including drift, noise, applying the Fujita-Lawler algorithm, etc) would show no difference between what corresponds to Fe1 and Fe2 sites.

We thank the reviewer for this suggestion. Following, we will show the validity of the algorithm by the simulated results.

Figure R2a is a simulated STM image with a thermal drift, which we used to test to validity of the Lawler-Fujita algorithm. The Fourier transform of the drifted STM image is shown in Fig. R2b. The Bragg peaks are blurred due to the perturbations brought by the drift. The image after applying the Lawler-Fujita algorithm is shown in Fig. R2c. The Bragg peaks shrink into sharp pixels after using this drift correction process, which indicates an ideal lattice after the drift correction process.

In Figs. R2e and R2i, we simulate tetragonal FeSe conductance maps without and with the [110]

electronic order, respectively. The intensity of Q_x and Q_y is equal for the conductance map without the [110] electronic order (R2f). In comparison, Q_x and Q_y 's intensity is not equal if the [110] electronic order is considered (R2j). In our simulated data, the inequivalent Bragg peaks of Q_x and Q_y are brought by the inequivalent Fe₁ and Fe₂ atoms. We intentionally choose a small weight of Fe atoms (with Fe/Se = 12%) to match the experimental data at 77 K. For images in Figs. R2e and R2i, we apply the same Lawler-Fujita algorithm to test the reliability of the code. The drift corrected conductance maps are shown in Figs. R2g and R2k, respectively. There is no difference between the images before and after applying the Lawler-Fujita algorithm (see Fig. R2g (Fig. R2k) and Fig. R2e (Fig. R2i)). The intensity of Q_x and Q_y after the drift correction is also the same as the unprocessed data, as shown in Figs. R2h and R2l. Since we simulate the ideal lattice (Figs. R2e and R2i), it is reasonable to see no change between the processed and unprocessed data. The simulated results in Fig. R2 indicate that the Lawler-Fujita algorithm used in our experiment is a reliable process regarding the thermal drift.

Fig. R2 **a** Simulated STM image with a thermal drift. **b** Fourier transform of **a**. **c** STM image after applying the Lawler-Fujita algorithm to **a**. **d** Fourier transform of **c**. **e** Simulated FeSe conductance map without the [110] electronic order. Inset shows the Fourier transformed conductance map of **e**. **f** Intensity of the Bragg peaks of Q_x and Q_y . The center of the Fourier transform has been shifted to a site of Se site. **g** Drift corrected conductance map of **e** after applying the Lawler-Fujita algorithm. **h** Intensity of the Bragg peaks of Q_x and Q_y after applying the Lawler-Fujita algorithm. **i-l** The same as **e-h**, but with simulation considering the [110] electronic order.

For drift-corrected data in Figs. R2g and R2k, we extract the sites of Fe₁ and Fe₂ and calculate the difference between Fe₁ and Fe₂ within each unit cell. The local [110] order calculated from Figs. R2g and R2k are shown in Figs. R3a and R3c, respectively. For the dataset without the [110]

electronic order, both positive and negative signs are observed (Fig. R3a). The nonzero local order is from the integer pixels of Fe sites selected by the algorithm, which can be owed to the algorithmic error. The local order distribution in Fig. R3a shows a Gaussian type, and the expected value is zero (Fig. R3b). For the dataset with the [110] electronic order, the local order shows an overall positive sign. Some negative signs are also observed due to the integer pixels of Fe sites selected by the algorithm. We intentionally chose a pixel density of 200 pixels \times 200 pixels per 20 nm \times 20 nm to simulate the lowest pixel density used in our measurements. A local order map with an overall positive sign can be obtained if we set a larger pixel density and Fe/Se weight ratio. The distribution of the local [110] electronic order in Fig. R3c is also a Gaussian type. Unlike that in Fig. R3b, the expected value of the local order with [110] electronic order is nonzero (Fig. R3d). The simulated conductance maps have been normalized to an expected value of 0.12 (a.u.), roughly equal to the mean conductance value at -100 meV for the dataset in Fig. 2 of the main text (see also the dI/dV curves in Supplementary Fig. 4). The value of the simulated [110] order in Fig. R3d is also consistent with that in Fig. 2f in the main text. The simulated data with the [110] electronic order indicates that a small order with a magnitude of 10^{-3} is detectable with our algorithm.

Fig. R3 **a** Local [110] electronic order calculated from the simulated conductance map in Fig. R2g. **b** Distribution of the local [110] electronic order in **a**. **c** Local [110] electronic order calculated from the simulated conductance map with the [110] electronic order (Fig. R2k). **d** Distribution of the local [110] electronic order in **c**.

We also simulate the influence of the long-range noise on our results. Figs. R4a and R4e are simulated conductance maps with long-range noise. The difference is that there is no [110] electronic order in Fig. R4a, while a finite [110] electronic order is considered in Fig. R4e. As a result, the intensity of the Q_x and Q_y is equal in Fig. R4b, while the Q_x and Q_y 's intensity are not equal in Fig. R4f. The long-range noise does not influence Q_x and Q_y 's intensity since they correspond to vectors with different wavelengths. The drift corrected conductance maps after applying the Lawler-Fujita algorithm to Figs. R4a and R4e are shown in Figs. R4c and R4g, respectively. The corresponding intensity of Q_x and Q_y remains no change after the drift correction process. The results in Fig. R4 indicates that the Lawler-Fujita algorithm is robust against the long-range noise.

Fig. R4 **a** Simulated FeSe conductance map without the [110] electronic order but with long-range noise. The noise to signal ratio is set to be noise/signal = 0.2. Inset shows the Fourier transformed image of the conductance map. **b** Intensity of the Bragg peaks of Q_x and Q_y . **c** The image after applying the Lawler-Fujita algorithm to **a**. Inset shows the Fourier transformed map of the drift corrected image. **d**. The intensity of the Bragg peaks of Q_x and Q_y after applying the Lawler-Fujita algorithm. **e-h** The same as **a-d**, but with additional [110] electronic order considered. The noise to signal ratio is also set to be noise/signal = 0.2.

Fig. R5 **a** Local [110] electronic order calculated from the simulated noisy conductance map in Fig. R3c. **b** Distribution of the local [110] order in **a**. **c** Local [110] electronic order calculated from the filtered conductance map. **d** Distribution of the local [110] electronic order in **c**. **e-h** The same as **a-d**, but with simulation considering the additional [110] electronic order.

For the datasets in Figs. R4c and R4g, we also extract the sites of Fe_1 and Fe_2 and calculate the intra-unit cell electronic order. The results are shown in Figs. R5a and R5e, respectively. Due to the perturbations bring by the noise, the local order in Fig. R5a shows periodic patterns. The local order

in Fig. R5e is still an overall positive sign. Some local order values turn to be negative compared to that in Fig. R3c, which is caused by the noise perturbations. The distribution of the local [110] order with noise is broader than that without the noise when comparing Fig. R5f (Fig. R5b) to Fig. R3d (Fig. R3b). Despite the noise, the expected value of the [110] order is roughly the same between the two cases with and without the noise. We use the same filter size as in our manuscript to filter out the noise signal in Figs. R4c and R4g. The local order calculated from the filtered conductance maps is shown in Figs. R5c and R5g. The noise effect is substantially removed since the filtered local order maps (Figs. R5c and R5g) are almost the same as those without the noise (Figs. R3a and R3c). The widths of the distribution of the local order after the filtration process are also narrowed. The local order maps and the distribution maps after filtering in Fig. R5 are quantitatively consistent with those in Fig. R3. Some subtle differences can be observed due to the convolution's edge effect during the filtration process. The local [110] electronic order's consistency before and after the filtration process indicates that our filtration process is reliable.

At the very least, before the manuscript is considered for publication, the authors should include the unprocessed real space differential conductance maps in the supplementary material, together with the conductance maps after processing.

As the reviewer suggested, we added the unprocessed and processed conductance maps in Supplementary Figure 3.

Reviewer #2 (Remarks to the Author):

I think the authors have addressed very well of my concerns about the size of the energy gap. They showed data acquired on several different areas with logarithmic scales, which convincingly establish the energy gap of ~ 80 meV. However, a correlation between the [110] electronic order (determined by STM) and Neel spin fluctuations (determined by INS) is not trivial. An explanation of how to rule out that two independent phenomena exhibit similar energy scale features in their own spectra is still missing. Once the authors clearly address this issue, I would support publication in Nature Communications.

We agree with the reviewer that it's important to rule out that these two phenomena only exhibit similar energy scale features. When we put two correlated phenomena together, it's possible that there is no real correlation between them. In our experiment, we make analog between the [110] electronic order and Neel spin fluctuations because of several simultaneous correlations: 1) They are all along the [110] direction of the Fe-Fe lattice. 2) They have similar energy gaps. 3) The [110] electronic order shows competing behavior with the [100] nematic order, which is reminiscent of the competing relationship between Neel and stripe spin fluctuations. The latter has been speculated to be the origin of the [100] nematic order. 4) The wavevector of the [110] electronic order is the same as that of the Neel spin fluctuations. Although the possibility that the two are unrelated still exists, this possibility would be relatively very small.

Reviewer #3 (Remarks to the Author):

The authors have amended the manuscript in response to my last comment in a satisfactory way.

As far as I can judge, the manuscript is acceptable in Nature Communications.

Small note: Instead of "Phenomenally" (p.11), maybe the authors would like to write "Phenomenologically"?

We thank reviewer#3 for the recommendation of publication of our manuscript. We amended this typo mistake in the main text.